# Heritable changes in chromatin contacts associated with transgenerational susceptibility to diet-induced insulin dysregulation and obesity

Richard C. Chang [1,5], Riann J. Egusquiza[2,5], Angélica Amorim Amato[1,3], Zhuorui Li[1], Alivia L. Dougherty[1], Kaitlin T. To[1], Michelle Avila [1], Erika M. Joloya [1], Hailey B. Wheeler[1], Angela Nguyen[1], Keiko Shioda[4], Junko Odajima[4], Michael S. Lawrence [4], Toshi Shioda [4,6] & Bruce Blumberg [1,6]

Effects of prenatal environmental exposures can be transmitted across generations through the germline without DNA mutations, an example of transgenerational epigenetic inheritance. Understanding how such inheritance occurs remains a central unanswered question in biology. Here we show that gestational exposure of mice to the environmental obesogen tributyltin produces heritable changes in chromatin interactions within the *Ide* gene encoding insulin-degrading enzyme in male primordial germ cells. These altered contacts persist through the F3 generation and are accompanied by reduced hepatic *Ide* expression, hyperinsulinemia, hyperglycemia and hyperleptinemia, resembling the phenotype of *Ide*-deficient mice that are predisposed to adult-onset, diet-induced obesity. The formation of new chromatin contacts, suppression of *Ide* expression, and associated metabolic phenotypes occurs only in males. These findings reveal a plausible molecular mechanism by which environmental exposures induce location-specific, three-dimensional changes in chromatin structure that transmit susceptibility to metabolic disorders to subsequent unexposed generations in mammals.

Evidence is accumulating that transgenerational epigenetic inheritance (TEI) may negatively impact the health of mammals after gestational exposure of their ancestors to toxic substances, but without direct exposure of the affected generations. Several mechanisms have been proposed to explain mammalian TEI[1,2]—namely, changes in DNA methylation[3], histone methylation[4], histone retention[5,6], or transmission of small noncoding RNA species to germ cells[7]. However, whereas significant epigenetic alterations have been observed in germ

cells or somatic cells in multiple generations after the initial exposure to toxicants, specific toxicant-induced epigenetic changes that persist across the post-exposure generations and explain the inherited phenotypes have not been clearly identified in the genome of the germline cells. For example, many studies, including ours[8], showed that genome-wide DNA methylation profiles were altered after ancestral chemical exposures, but whether the persistent presence of such differentially methylated regions—which are apparently resistant to the

[1]Department of Developmental and Cell Biology, University of California Irvine, Irvine, CA, USA. [2]Department of Pharmaceutical Sciences, University of California Irvine, Irvine, CA, USA. [3]Department of Pharmaceutical Sciences, University of Brasília, Brasília, Brazil. [4]Krantz Family Center for Cancer Research, Massachusetts General Hospital, 149 13th Street, Charlestown, MA, USA. [5]These authors contributed equally: Richard C. Chang, Riann J. Egusquiza. [6]These authors jointly supervised this work: Toshi Shioda, Bruce Blumberg. ✉e-mail: shioda@mgh.harvard.edu; blumberg@uci.edu

two waves of global DNA demethylation events during the early phase of mammalian germ cell development—is required for TEI or not remains to be determined[8,9]. Contributions of histone modifications or noncoding RNA species to TEI also await identification of specific and persistent epigenetic changes that are directly responsible for the inherited phenotypes[10].

Our preceding studies suggested that stable alterations of higher-order chromatin structures might provide a unifying theory to explain mammalian transgenerational inheritance[8,11]. Support for such a model requires the identification of persistently altered regions of higher-order chromatin structure that are passed to subsequent generations. Therefore, we sought to identify regions of persistently altered chromatin interactions (CIs) in PGCs from mice exposed to the environmental obesogen tributyltin (TBT) (F1) and unexposed (F2, F3) generations and associate these specific changes with the transgenerational susceptibility to obesity phenotypes observed. These alterations in CIs are likely to represent changes in higher-order chromatin structure, that can be inferred from the changed CIs.

Here, we used an established, highly reproducible murine model of male-specific transgenerational susceptibility to diet-induced obesity to show that exposure to TBT elicited heritable changes in chromatin interactions (CIs) in primordial germ cells (PGCs) and that such CIs may contribute to the transgenerational metabolic phenotypes. New CIs were formed within the *Ide* gene encoding insulin-degrading enzyme in the directly exposed PGCs, then stably maintained in PGCs of the subsequent (unexposed) two generations. Concomitantly, *Ide* mRNA expression was decreased in the livers of male descendants from the exposed dams. These males were hyperinsulinemic and hyperglycemic, phenocopying *Ide*-deficient mice that are predisposed to adult-onset, diet-induced obesity. Our results provide a plausible molecular mechanism underlying the TEI of male-specific predisposition to obesity caused by gestational exposure of mice to TBT, a representative environmental obesogen. They also provide an entry point for future studies aimed at understanding how environmental exposures can cause location-specific changes in the 3D chromatin structure to influence physiology across multiple generations in mammals.

## Results

### Ancestral TBT exposure led to a transgenerational predisposition to increased WAT mass

In the new transgenerational experiment 4 (T4) detailed here, effects of ancestral TBT exposure throughout gestation were confirmed to be much stronger in male versus female F2 and F3 generation C57BL/6J mice. This confirms our previous transgenerational experiments denoted as T1[12], T2[8], and T3[13] and another study using OG2 C57BL/6J mice[14]. Salient findings included increased WAT depot size, more overall body fat and in some cases, increased body weight (Fig. 1). After the diet was changed from a standard chow diet (SD) to a higher fat diet (HFD) at 5 weeks of age, only at 11 weeks did male F2 animals ancestrally exposed to TBT accumulate significantly more WAT than controls (Fig. 1a); body weight did not differ between groups. In contrast, F3 male animals did not show increased body weight or fat mass when switched to the HFD at 5 weeks (Supplementary Fig. 1). We hypothesized that HFD challenge had started too soon compared with previous experiments[8,13]. Therefore, we switched sibling DMSO- and TBT-group F3 animals that had been maintained on the SD to HFD at 17 weeks of age. TBT-group F3 males rapidly accumulated body weight and fat mass compared with controls; these differences became statistically significant at 19 (body fat) or 20 weeks (body weight) (Fig. 1b). No effects of ancestral TBT exposure on fat accumulation were observed in females (Fig. 1c, d).

### Ancestral TBT exposure caused transgenerationally stable changes in chromatin interactions in the genome of male PGCs

To determine whether CIs were stably altered after ancestral TBT exposure, we performed Hi-C seq of PGCs isolated from E13.5 embryonic gonads of F1–F3 mice, pooled by litter and sex. Approximately 20,000 FACS-enriched PGCs from each group were subjected to Hi-C sequencing, data generation and analysis (Supplementary Fig. 2). Successful detection of a known set of TADs around the *HoxD* gene cluster[15] demonstrated the validity of our Hi-C seq data (Supplementary Fig. 2i, j).

PGCs in F1 embryos were directly exposed to TBT while the embryos were within the treated F0 dams. Those isolated from F2 or F3 embryos were not exposed. PGCs in F2 embryos became gametes producing F3 animals, which showed a transgenerational predisposition to diet-induced obesity[8,12–14]. The global profiles of chromatin contacts were well conserved in PGC genomes across sex, F0 exposure to TBT, or F1–F3 generations (Supplementary Fig. 3a). The median distance between two chromatin contacts was ~1 megabase (Supplementary Fig. 3b), which agrees with the previously reported size of TADs in the mouse genome (880 kb)[16].

Applying a strict criterion of detection (differential chromatin interaction scores >3.0), we identified 20 autosomal differential chromatin interactions (DCIs) conserved between F1 and F2 male PGCs and only one DCI in chromosome 19 that was conserved across all generations (F1–F3) of male PGCs (Supplementary Fig. 3c).

The DCI score plots shown in Fig. 2a demonstrate a region containing significant DCIs—which were gained in the TBT group compared to the DMSO group—well conserved in chromosome 19 of male PGCs across F1–F3 generations (chr19:36,920,000-37,420,000; blue horizontal bar) but not in female PGCs. Formation of new CIs in this region in the genome of the TBT-group PGCs was confirmed by direct visual inspection of normalized chromatin contact matrix data of male PGCs isolated from F1–F3 embryonic testes although these new contacts became weaker in F3 embryonic testes (Fig. 2b). In contrast, there were no changes in CIs in this region among female PGCs across generations after ancestral TBT exposure compared with vehicle controls (Supplementary Fig. 4a). Visual inspection of DCIs for the whole chromosome 19 of male PGCs isolated from the F2 embryos identified only a single region displaying discernible DCIs (Supplementary Fig. 4b, top panel), which was confirmed to be identical to the region described above (Supplementary Fig. 4b, zoom-in panels). Detailed examination of this region revealed three DCIs— namely, two small DCIs around chr19:36,920,000-37,020,000 (DCI1) and chr19:37,220,000-37,420,000 (DCI2) and a large DCI (DCI-3) spanning over the two small DCIs (Fig. 2c and Supplementary Fig. 4b). The nested structure involving these three DCIs convincingly supports TBT-induced formation and transgenerational persistence of these DCIs. To increase our confidence that the two small DCIs (DCI-1 and DCI2) involved in the longer-range DCI (DCI-3) were not problematic genomic regions, we confirmed that DCI1-3 are not flagged by the ENCODE blacklist[17], which are genomic regions unsuitable for deep sequencing-based analyses due to unusual structures. We also confirmed that DCI1-3 are not involved in any segmental duplication as detected by SDquest[18], eliminating the possibility that DCI1 and DCI2 are deep sequencing artifacts stemming from significantly similar sequences locating nearby. Interestingly, DCI2 contained the *Ide* gene, which encodes insulin-degrading enzyme (Fig. 2b, c). Hepatic *Ide* is responsible for the majority of insulin clearance[19]. These results demonstrated the formation of transgenerational, germline-transmitted alterations in CIs after exposure of pregnant F0 female mice to the obesogen, TBT.

### Ancestral TBT exposure induced CI formation in the *Ide* gene in male livers

To determine whether the DCIs identified in PGCs were also found in the *Ide* gene in somatic tissues, we sought to examine CTCF binding status in F3 livers because liver is the primary site of *Ide* expression and F2 PGCs give rise to F3 descendants. Five CTCF binding sites predicted by the JASPAR transcription factor binding site database lie near the CI

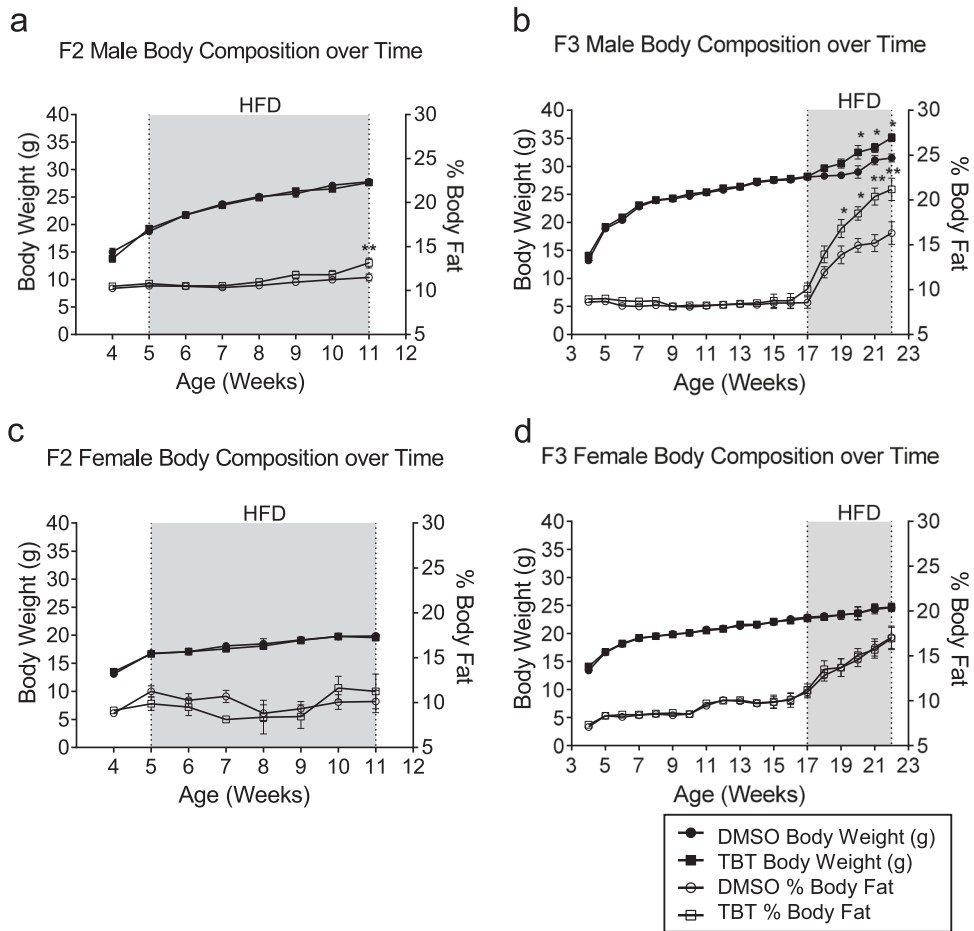

**Fig. 1 | Mice ancestrally exposed to TBT exhibited increased fat content in F2 and F3 male descendants.** Body weight and relative body composition of **a** F2 male descendant (*n* = 15), **b** F3 male descendant (*n* = 16), **c** F2 female descendants (*n* = 17), and **d** F3 female descendants (*n* = 16) throughout the time course of the experiment. The grey area indicates the period of the diet challenge. Statistical significance was determined using two-way ANOVA. Pair-wise Bonferroni post hoc tests were used to compare different groups ($p = 0.009$ for F2 TBT males at week 11; $p = 0.038$ for F3 DMSO males at week 20; $p = 0.021$ for F3 DMSO males at week 21; $p = 0.032$ for F3 DMSO males at week 22; $p = 0.041$ for F3 TBT males at week 19; $p = 0.037$ for F3 TBT males at week 20; $p = 0.001$ for F3 TBT males at week 21; $p = 0.004$ for F3 TBT males at week 22;). Data were presented as mean ± s.e.m. \**p* < 0.05; \*\**p* < 0.01. Each data point represents an independent biological replicate (individual mouse).

(A–E) together with two known CTCF binding sites (F, G) (Chr19:37,320,000 in mm10) (Fig. 3a). ChIP-qPCR analysis was performed on CTCF or RAD21 chromatin pulldown samples. We observed increased CTCF binding in four regions (B, C, F, G) within the *Ide* gene in male livers and infer that these may form a small chromatin loop (Fig. 3b). These findings were reproduced with different primer sets (Supplementary Fig. 5). No significant enrichment was noted in TBT female samples (Fig. 3c). We did not detect enrichment of RAD21 binding on the *Ide* gene (Fig. 3d, e).

We assessed the CTCF binding profile of the whole chromosome 19 (61.7 Mb) using ChIP-seq and found that it was largely identical between the DMSO and TBT groups of the F3 male livers (Supplementary Fig. 6a). Exposure-specific differences were not readily detected in 17.6 Mb or 4.4 Mb windows around the *Tnks2-Myof* genes, which did not contain very strong CTCF binding sites (Supplementary Fig. 7b, c). However, focused analysis of a ~1 Mb region spanning from *Tnks2* to *Exo6* revealed differential CTCF binding sites (DCBSs) as well as preserved binding sites between the DMSO and TBT groups (Supplementary Fig. 6a). One DCBS observed near the first exon of the *Btaf1* gene was greater in the DMSO group than the TBT group (Supplementary Figs. 6a and 7d–f; blue wedge) whereas three DCBSs within the *Cpeb3* gene were greater in the TBT group than the DMSO group (Supplementary Figs. 6 and 7d, f red wedges). Seven relatively strong CTCF binding sites were preserved

between the DMSO and TBT groups (Supplementary Figs. 6 and 7d–h). The intra-*Cpeb3* DCBSs, which were stronger in the TBT group than the DMSO group, may contribute to the centromeric (left-side) boundary of the dTAD that contains the whole *Ide* gene (Supplementary Fig. 6b, red triangle) while the preserved CTCF binding sites near the *Hhex* gene may form the other (telomeric, right-side) boundary (Supplementary Figs. 6 and 7d–g). The DCBS near Exon I of *Btaf1* is in the close vicinity of a well-preserved CTCF binding site at the *Fgfbp3* gene (Supplementary Figs. 6 and 7d–f). It is tempting to speculate that CTCF bound to the *Btaf1* site may interact with CTCF bound to the *Fgfbp3* site in the liver of normal adult males, blocking the accessibility of the *Fgfbp3* site from interacting with other CTCF. When CTCF binding to the *Btaf1* site is lost by the ancestral exposure to TBT, CTCF at the *Fgfbp3* site may be released and can interact with the CTCF bound to the *Hhex* area, forming a large and new dTAD (Supplementary Fig. 6b, blue triangle). To elucidate the molecular mechanisms regulating *Ide* gene expression in adult mouse liver and the effects of the ancestral exposure to TBT, future studies will be necessary to characterize detailed protein-protein interactions between the CTCF proteins bound to the DNA regions identified in the current study. The binding of CTCF within the *Ide* gene was not detected by our ChIP-seq. Presumably, these interactions were too weak for detection by ChIP-seq, although ChIP-PCR had sufficient sensitivity to quantify them.

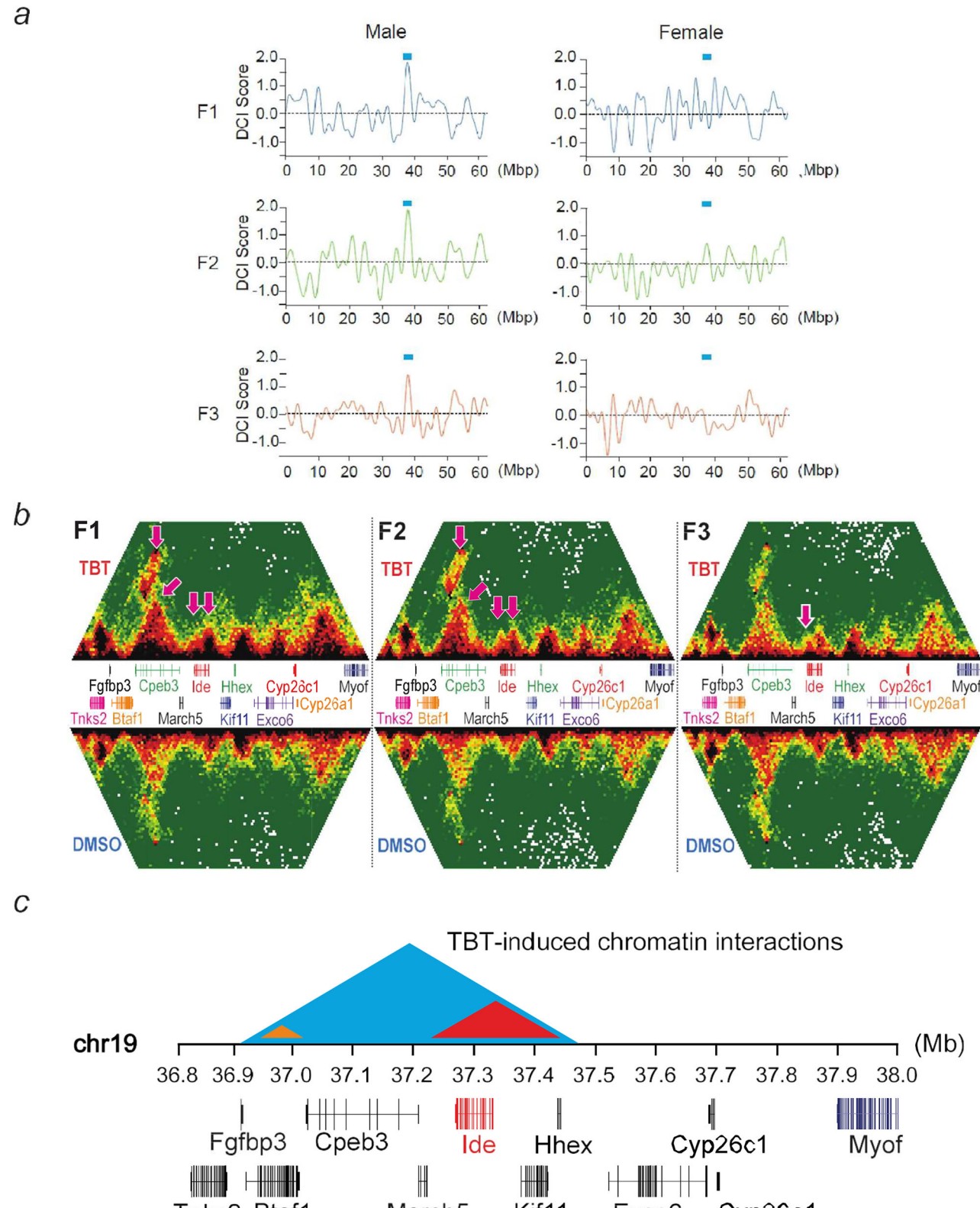

**Fig. 2 | Transgenerationally transmitted differential chromatin interactions (DCI). a** DCI scores of the whole chromatin 19 in mouse primordial germ cells isolated from F1 to F3 embryos were determined by Bart-3D and smoothened for plotting. Blue horizontal bars indicate the location of DCI at the *Ide* gene. **b** Chromatin contact plots of primordial germ cells isolated from F1 to F3 male embryos after F0 exposure to TBT (top) or DMSO (bottom). Arrows indicate DCIs gained by the F0 exposure to TBT. **c** Locations of genes near the transgenerationally conserved DCIs caused by F0 exposure to TBT. The Ide gene is shown in red. PGCs were isolated from E13.5 gonads of embryos derived from five independent dams per generation. Gonads within each litter were pooled by sex before sorting, and the resulting male and female PGC fractions were used for chromatin interaction analysis.

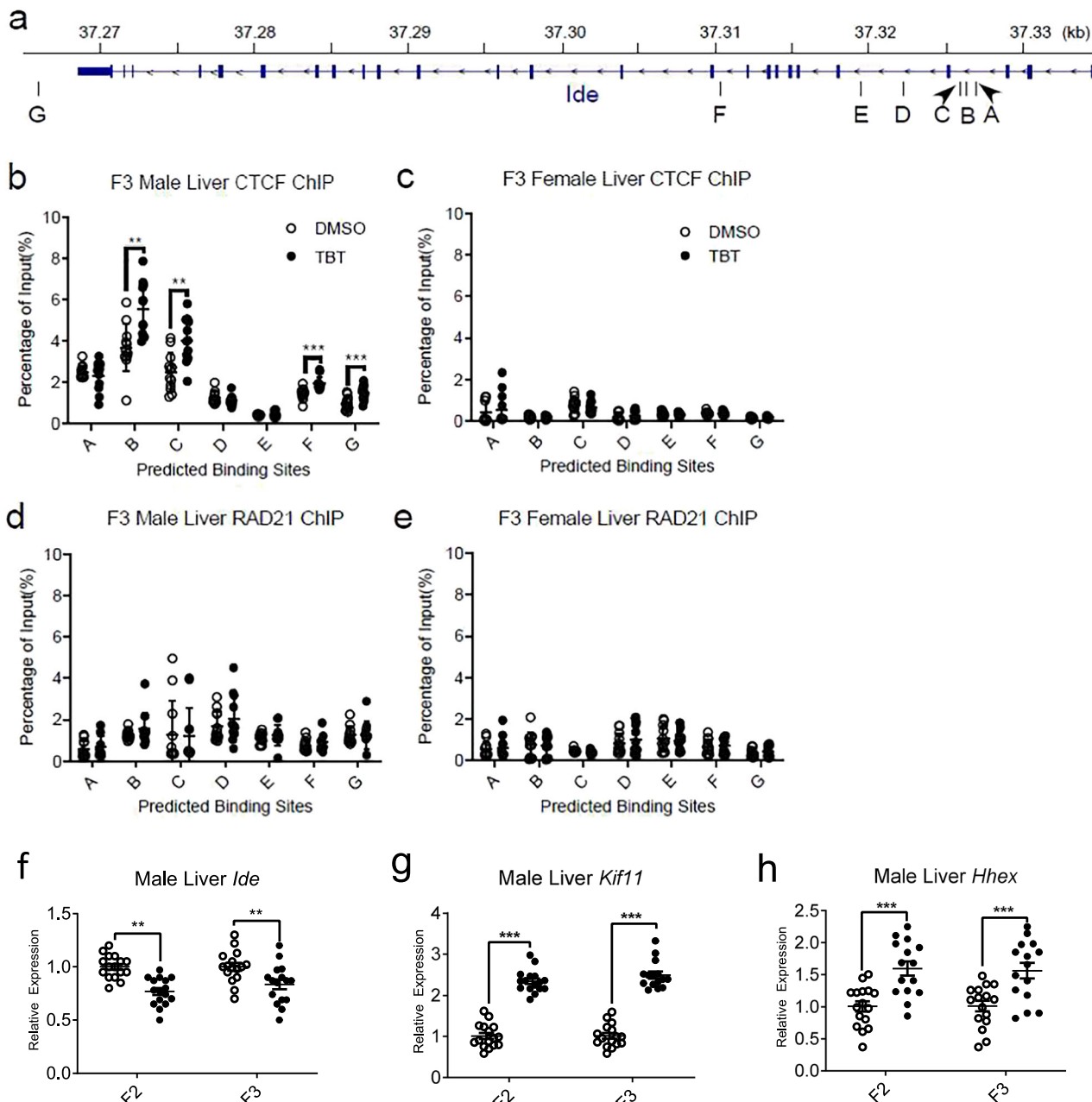

**Fig. 3 | Mice ancestrally exposed to TBT showed increased CTCF binding at the *Ide* gene on chromosome 19, accompanied by altered expression of genes encoding *Ide, Hhex,* and *Kif11*. a** Five potential CTCF binding sites (A to E) in the *Ide* gene of chromosome 19 predicted by JASPAR on the UCSC Genome Browser and two known CTCF binding sites (F and G) were analyzed. Chromatin immunoprecipitation (ChIP) and quantitative real-time RT-PCR (qPCR) assays using an antibody against CTCF in F3 **b** male or **c** female descendants' livers ($n = 12$). ChIP and qPCR assays using an antibody against RAD21 in F3 **d** male or **e** female descendants' livers. Normal rabbit IgG was used as a non-specific antibody control. Unpaired $t$-tests were used for qPCR analysis. ($p = 0.007$ for male predicted CTCF binding site

B; $p = 0.009$ for male predicted CTCF binding site C; $p < 0.001$ for male predicted CTCF binding site F; $p < 0.001$ for male predicted CTCF binding site G). Data were presented as mean ± s.e.m. *$p < 0.05$; **$p < 0.01$; ***$p < 0.001$. Expression of *Ide, Kif11,* and *Hhex* in male livers. The relative mRNA levels of the **f** *Ide*, **g** *Hhex*, and **h** *Kif11* genes were assayed by quantitative PCR of F2 ($n = 17$) and F3 ($n = 16$) male descendants from the current experiment. Data were expressed as mean fold change ± s.e.m. and assayed in duplicate. Significance was assessed by unpaired $t$-test versus DMSO controls ($p = 0.008$ for F2 *Ide*; $p = 0.009$ for F3 *Ide*; $p < 0.001$ for both F2 and F3 *Kif11*; $p < 0.001$ for both F2 and F3 *Hhex*).

## Ancestral TBT exposure led to a transgenerational, male-specific hyperinsulinemia

Formation of transgenerationally persistent, novel CIs in male PGCs within the *Ide* gene (Fig. 2 and Supplementary Figs. 3 and 4) prompted us to hypothesize that expression of *Ide* and neighboring genes *Kif11* and *Hhex* mRNAs might be affected by ancestral TBT exposure. Strikingly, significant decreases in *Ide* mRNA expression were found in

the livers of F2 and F3 adult males (Fig. 3f) but not in skeletal muscle (Supplementary Fig. 8a, gonadal WAT (Supplementary Fig. 8b), spleen, or brain (Supplementary Fig. 21a, b) or in females (Supplementary Fig. 8g, j, m). Notably, *Ide* mRNA expression was also decreased in F3 male livers from the previously published T3 experiment[13] (Supplementary Fig. 9), indicating that this phenotype is reproducible across experiments. Strikingly, expression of *Kif11* and *Hhex* mRNAs was up-

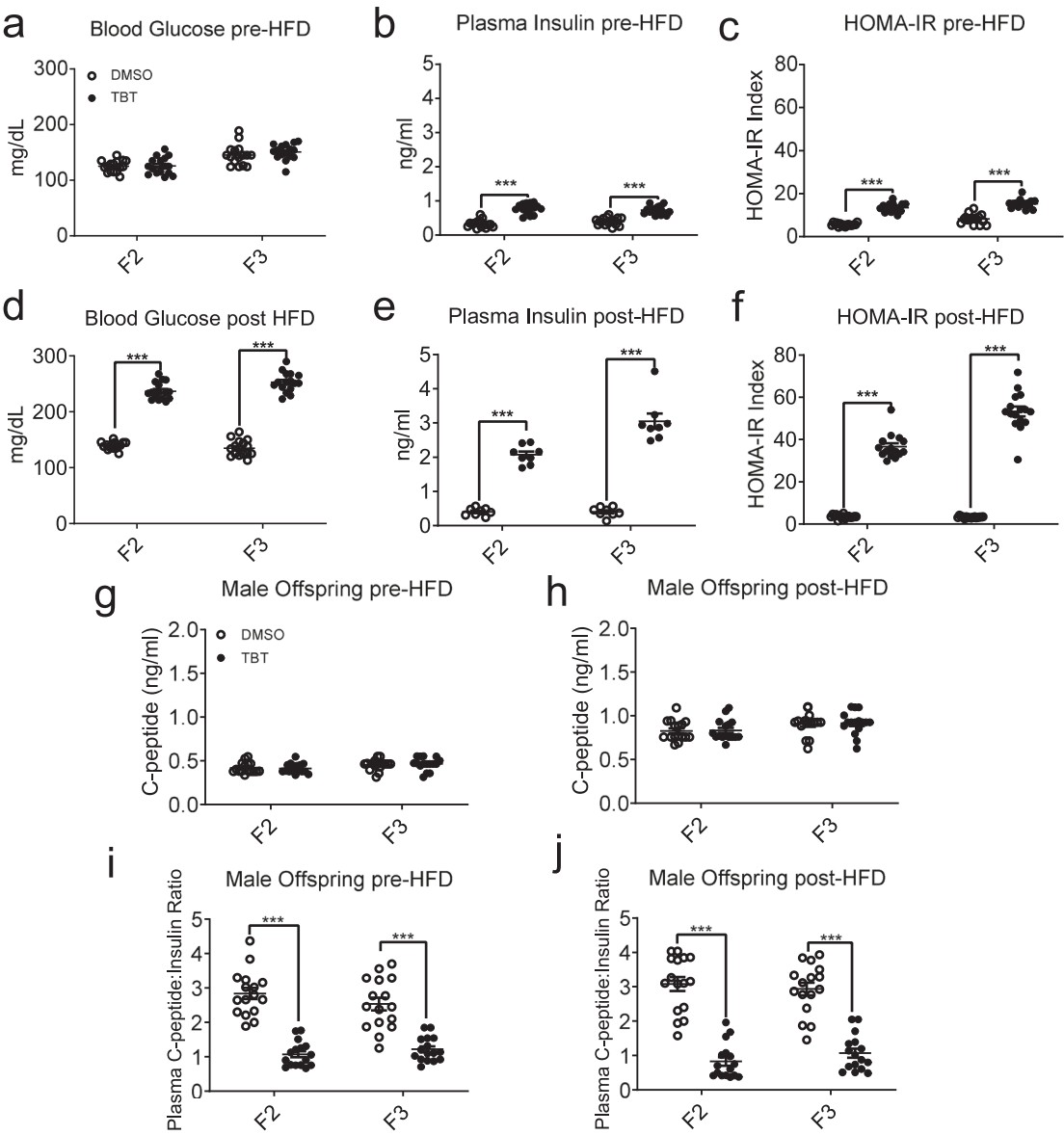

**Fig. 4 | Male mice ancestrally exposed to TBT suffer from hyperinsulinemia before and after 6 weeks of diet challenge that results from impaired insulin clearance. a** Plasma glucose, **b** insulin of male mice before diet challenge (F2 at 5 weeks age and F3 at 17 weeks age) were measured, and **c** HOMA-IR indexes were calculated. **d** Plasma glucose and **e** insulin of male mice after 6 weeks of diet challenge (F2 at 11 weeks age and F3 at 22 weeks age) were measured and **f** HOMA-IR indexes were calculated. Plasma C-peptide concentrations (F2 at 5 weeks age and F3 at 17 weeks age) were measured in (**g**), male mice before and **h** after diet challenge. Plasma C-peptide and insulin (measured in moles/L) were measured and the ratio of C-peptide:insulin calculated and presented before **i**, (F2 at 5 weeks, F3 at 17 weeks) and after **j**, (F2 at 11 weeks, F3 at 22 weeks) HFD diet challenge. Statistical significance was determined using two-way ANOVA followed by Bonferroni post hoc tests ($p < 0.001$ for both F2 and F3 plasm insulin pre-HFD; $p < 0.001$ for both F2 and F3 HOMA-IR pre-HFD; $p < 0.001$ for both F2 and F3 plasm glucose post-HFD; $p < 0.001$ for both F2 and F3 plasm insulin post-HFD; $p < 0.001$ for both F2 and F3 HOMA-IR post-HFD; $p < 0.001$ for both F2 and F3 C-peptide: insulin ratio pre-HFD; $p < 0.001$ for both F2 and F3 C-peptide: insulin ratio post-HFD). Data were presented as mean ± s.e.m. *$p < 0.05$; **$p < 0.01$; ***$p < 0.001$. Each data point represents an independent biological replicate (individual mouse). F2, $n = 16$; F3, $n = 17$.

regulated in male livers (Fig. 3g, h) but not in skeletal muscle (Supplementary Fig. 8b, c), gWAT (Supplementary Fig. 8e, f) or in females (Supplementary Fig. 8h, I, k, l, n, o). In contrast, the expression of genes upstream of *Ide* such as *March5*, *Cpeb3*, *Btaf1*, and *Fgfbp3* was largely unchanged in either males (Supplementary Fig. 19) or females (Supplementary Fig. 20). There was a slight increase in expression of hepatic *Cpeb3* expression in males (Supplementary Fig. 19d) whereas expression of *Btaf1* in male gWAT was slightly decreased (Supplementary Fig. 19h). These data are consistent with a model in which the altered TAD structures in and around *Ide* lead to decreased *Ide* expression accompanied by increased *Hhex* and *Kif11* expression in liver.

No differences in fasting blood glucose levels were observed in either males (Fig. 4a) or females (Supplementary Fig. 10a) prior to diet challenge, as we previously reported[8]. However, plasma insulin levels in TBT group males were significantly higher (Fig. 4b), in accord with the decreased hepatic *Ide* mRNA expression (Fig. 3f) whereas no significant differences were observed in females (Supplementary Fig. 10b). We calculated homeostasis model assessment of insulin resistance (HOMA-IR) to estimate insulin resistance and found that TBT-group male descendants showed higher potential for insulin resistance than did DMSO descendants (Fig. 4c). Females showed no differences (Supplementary Fig. 10c). After 6 weeks of HFD diet, TBT male descendants were hyperglycemic (Fig. 4d) and hyperinsulinemic (Fig. 4e),

in addition to having increased fat content (Fig. 1a, b). Females showed no effects (Supplementary Fig. 10d, e). HOMA-IR index was strikingly increased in the male (Fig. 4f) but not female (Supplementary Fig. 10f) TBT animals, suggesting a high likelihood of insulin resistance compared with vehicle controls.

To address whether insulin secretion or insulin breakdown was responsible for increased insulin levels in TBT-group males, we measured C-peptide (Fig. 4g, h) and calculated C-peptide:insulin ratios (Fig. 4i, j). Analysis of C-peptide levels revealed that there were no differences between DMSO and TBT group males (Fig. 4g) or females before HFD challenge (Supplementary Fig. 10g), or females after HFD challenge (Supplementary Fig. 10h). C-peptide levels increased ~50% in DMSO and TBT-group males after HFD challenge, indicating that HFD led to increased insulin secretion (Fig. 4h). Measurement of the ratios of C-peptide:insulin indicated that increased insulin levels observed in TBT-group males before and after HFD were likely the result of impaired insulin clearance resulting from decreased IDE production (Fig. 4i, j). Females were unaffected (Supplementary Fig. 10i, j). In accord with our previous publications[8,13], plasma leptin levels were strongly increased in TBT-group males (Supplementary Fig. 11a), but not females, after HFD challenge (Supplementary Fig. 11b).

## Discussion

Obesogens are chemicals that lead to increased WAT mass in exposed organisms[20,21]. TBT activated the nuclear receptors peroxisome proliferator activated receptor gamma (PPARγ) and its heterodimeric partner, retinoid "X" receptor (RXR)[22,23], leading to increased commitment of multipotent mesenchymal stromal stem cells to the adipose lineage and differentiation of pre-adipocytes into mature adipocytes[24]. TBT promoted fat accumulation in vivo across a variety of model systems, including mice[20,21]. Our previous experiments demonstrated increased WAT accumulation when dietary fat was elevated modestly (21.6 vs 13.1% calories from fat) in male F3 and F4 descendants of pregnant F0 mouse dams exposed to TBT throughout pregnancy[12,13] or pregnancy and lactation[8,11,14]. TBT exposure of pregnant F0 dams resulted in a stable, male-specific predisposition to obesity in exposed (F1 were exposed in utero, F2 were exposed as germ cells in F1) and unexposed (F3, F4) descendants.

We previously identified blocks of iso-directional, differentially methylated DNA (isoDMBs) in WAT of F4-generation male mice after exposure of F0 dams to TBT throughout pregnancy and lactation[8,11]. Genomic DNA regions in WAT where isoDMBs were under-methylated compared to controls were enriched in metabolic genes such as leptin, and these regions were less accessible in F3 and F4 sperm of the TBT group than in controls[8]. We proposed that ancestral TBT exposure caused changes in higher-order chromatin structure that were then inherited or reconstructed every generation, ultimately resulting in changes in chromatin accessibility and DNA methylation that altered expression of adipogenic and metabolic genes compared with controls[8,11].

The concept of mammalian TEI challenges the widely accepted view that changes in the DNA nucleotide base sequence are the exclusive basis of inheritance of traits acquired in one generation. To explain mammalian TEI, several epigenetic mechanisms have been proposed. However, because distinct types of epigenetic marks, such as DNA methylation, histone modifications, or noncoding RNA species, affect each other—which is known as epigenetic crosstalk—it is conceivable that the epigenetic aberration directly created by an environmental toxicant may not be identical to epigenetic alterations causing the phenotypes.

Transgenerational inheritance of altered CIs, as described in our current study, offers an attractive model in which multiple types of epigenetic alterations in a relatively large, but still specific region in the genome can be inherited in a coordinated fashion. An aberrant 3D structure in a chromosome can affect various distinct layers of the epigenetic landscape (e.g., DNA methylation, histone modification, histone retention, and ncRNA expression) in the affected region. In turn, anomalies in certain epigenetic machineries, such as DNA methylation-sensitive recruitment of CTCF, can modify 3D chromatin structure. Thus, the CI-based mechanism of TEI may provide clues to understand how the initial epigenetic aberration directly introduced by an environmental exposure can be converted to heritable, phenotype-causing epigenetic aberrations in the context of localized epigenetic crosstalk. Higher-order chromatin structure is often reflected by the presence or absence of chromatin TADs and loops that modulate accessibility to DNA and histone-modifying enzymes, to histones and to the transcription machinery[16]. Support for such a model required the identification of persistently altered regions of higher-order chromatin structure.

Here, we used Hi-C-seq analysis to identify CIs whose presence was stably altered in PGCs by direct or ancestral TBT exposure. Critically, the most high-scoring CI identified objectively was on chromosome 19 within the *Ide* gene. While we have only assessed the presence of CIs in PGCs from the current (T4) experiment, expression of *Ide* mRNA was also reduced in livers from a previous experiment (T3)[13]. Reduced *Ide* expression did not affect fasting glucose levels in the current T4 experiment, but significantly altered basal insulin levels, then led to strong increases in both glucose and insulin levels in HFD-challenged F2 and F3 males. This increase in insulin was not the result of increased insulin secretion because C-peptide levels did not change between the vehicle and TBT groups. Since only the male animals in our transgenerational experiments responded to diet challenge and then only increased WAT mass after HFD diet was initiated, decreased hepatic *Ide* expression in males appears to be a strong component of the transgenerational susceptibility to obesity. Previous gene knockout studies confirmed a role for IDE in insulin clearance; *Ide* loss-of-function produced hyperinsulinemia and age-dependent glucose-intolerance[19]. It is also notable that leptin levels were increased in the TBT-group males since it is known that hyperinsulinemia and insulin resistance impair leptin signaling, leading to leptin resistance[25]. Our findings are consistent with a mechanism involving altered hepatic *Ide* expression, and the fact that the observed downregulation of *Ide* phenocopies the effects of *Ide* loss-of-function supports a central role for *Ide* in the phenotypes observed. However, these results do not rule out the potential contributions of *Cpeb3* or other TAD-embedded genes, in other metabolic tissues. Future studies are warranted to investigate the role of these genes in metabolic processes and their potential interactions with environmental exposures.

Together with our previous studies[8,11], the new data presented here support a model in which transgenerational, non-genetic propagation of environmentally-induced phenotypes relied on alterations in chromatin structure. Consistent with our model, Corces and colleagues found that recruitment of CTCF to an enhancer in the obesity associated gene, *Fto* was associated with transgenerational inheritance of obesity after prenatal bisphenol A exposure[26,27]. There is also evidence that CTCF binds genomic DNA in sperm to form chromatin contacts, which may function as vehicles of paternally transmitted epigenetic inheritance[28].

We are aware of the possibility that the phenotypes observed in the F2 and F3 male mice might be created by the same type of phenotypes in the preceding generation without involving heritable materials relevant to the genetic or epigenetic mechanisms. For example, if obesity during pregnancy affects the metabolism of her direct children in a way that causes adult-onset obesity, the phenotype could appear in multiple generations without involving inheritance through germ cells. However, in our study, the metabolic phenotypes were observed only in male animals descended from TBT-treated dams, the pregnant dams were not subjected to HFD in any generation and no evidence of metabolic impairments during pregnancy of females were observed. Therefore, while we cannot currently

completely exclude the possibility, we do not consider a hypothetical mechanism of repeatedly created obesity as a primary mechanism underlying our observations.

It was recently called to our attention that a 0.6-Mb region in the genome of the C57BL/6J strain surrounding the *Ide* gene harbors a copy number variation (CNV) due to localized genome duplication, although the penetrance of this anomaly is variable (but ~50–60%) in C57BL/6 J[29] populations. While we note that our Hi-C analysis cannot completely distinguish between increased signals from bona fide TADs and increased signals from the presence of CNVs, we believe it is unlikely that the results presented here result from said CNVs. First, Watkins-Chow and Pavan reported that the extra copy of *Ide* and *Fgfbp3* present in the CNV led to ~1.5-fold increased levels of *Ide* mRNA in brain and spleen and 1.5-fold increase in *Fgfbp3* mRNA in spleen but not brain[29]. In contrast, we found that *Ide* expression was significantly decreased in livers of TBT group male mice, but unchanged in other tissues or in females (Supplementary Figs. 19 and 20). Second, we also evaluated expression of *Ide* and *Ffgbp3* in brain and spleen from a repeated transgenerational experiment[30] where the same putative TADs were identified and found no changes in *Ide* or *Fgfbp3* expression (Supplementary Fig. 21). Third, we identified increased binding of CTCF to four sites but not to three other sites in the *Ide* gene (Fig. 3 and Supplementary Fig. 5) of male, but not female TBT-group animals. This is contrary to what would be expected if the dTAD and apparent increase in CTCF binding only resulted from a CNV. Fourth, the CNV in *Ide* was reported to be present in only some mice from the Jackson Labs colony[29]. We randomized all of the mice obtained for this experiment prior to breeding and treated 74 F0 females with DMSO vehicle and 74 with TBT. Subsequent generations used groups of at least 61 (F1) and 69 (F2) females for breeding the next generations. While it might be possible that only the TBT group received mice harboring the putative CNVs, it is inconceivable that only the male offspring of these mice received this autosomal CNV compared with their female littermates, which showed no evidence of the dTAD or increased CTCF binding. Analysis of *Ide* CNV showed that there were no significant differences in *Ide* copy number between DMSO control group and TBT group males or females (Supplementary Fig. 22). Most critically, we compared *Ide* expression with *Ide* copy number from livers of the same animals and found no relationship between *Ide* copy number and *Ide* expression in among groups (Supplementary Fig. 23). *Ide* expression was strongly reduced in TBT-group males (but not females) at similar *Ide* copy number to DMSO group animals. Taken together, we consider that our results argue strongly against the presence or inheritance of CNVs in and near *Ide* being responsible for transmitting the transgenerational phenotype; although, our study does not completely exclude possible roles of the CNVs in formation of the epigenetic anomalies.

Altered chromatin structure necessarily changes the accessibility of DNA and histones to modifying enzymes such as DNA and histone methyl transferases, the location and retention of histones and the expression of various genes, including those for noncoding RNAs. These mechanisms could interact and be preserved across generations and in various types of differentiated cells[11]. Whether the observed, TBT-induced changes in chromatin contacts around the *Ide* gene in primordial germ cells originated from altered CTCF binding in sperm, or such changes are preserved in the sperm genome, are interesting questions. Future studies aimed at providing a cause-effect relationship between altered chromatin contacts/structure and the transgenerational inheritance of the effects of environmental exposures will be important to establish transgenerational epigenetic inheritance as an important mechanism underlying phenotypic change in response to environmental perturbations in mammals.

## Methods

### Chemicals and reagents

TBT, dexamethasone, isobutylmethylxanthine, and insulin were purchased from Sigma-Aldrich (St. Louis, MO). Rosiglitazone (ROSI) was purchased from Cayman Chemical (Ann Arbor, MI). Embryoid Body Dissociation Kit (#130-096-348) was purchased from Miltenyi Biotec (North Rhine-Westphalia, Germany). Zombie Red Dye (#77475) was purchased from BioLegend (San Diego, CA). PE Mouse anti-SSEA-1 (#560142) and Alexa-Fluor 647 Mouse anti-CD61 (#563523) were purchased from BD Biosciences (Franklin Lakes, NJ). Blood glucose meter kits (BG1000) were purchased from Clarity Diagnostics (Boca Raton, FL). Mouse Leptin ELISA Kit (#90030), Mouse C-peptide ELISA kit (#80954) and mouse insulin ELISA kit were purchased from Crystal Chem (Elk Grove Village, IL, USA). Arima-Hi-C Kit (A510008) was purchased from Arima Genomics (San Diego, CA). Ultra-pure formaldehyde (#18508) was purchased from Ted Pella Inc. (Redding, CA). MinElute Reaction Cleanup Kit (#28206) was purchased from Qiagen (Hilden, Germany).

### Animal maintenance and exposure

C57BL/6J mice were purchased from the Jackson Laboratory (Sacramento, CA) and housed in micro-isolator cages in a temperature-controlled room (26 °C) with a 12 h light/dark cycle. Water and food were provided ad libitum unless otherwise indicated. Animals were treated humanely and with regard for the alleviation of suffering. At the moment of euthanasia, each mouse was assigned a code, known only to a lab member not involved in the dissection process. All tissue harvesting was performed with the dissector blinded to which groups the animals belonged. Group sizes were based on our prior experiments and a prior power analysis

For this new transgenerational experiment, denoted as T4, we purchased 50 male and 148 female C57BL/6J mice (5 weeks of age). The number of mice used was based on a priori power analysis aimed at detecting a 10% difference in body fat with sufficient statistical power. We also have a stringent selection criterion for litter size, requiring litters of 6–8 pups with at least two of each sex (one for phenotypic analysis and another for breeding the next generation). Female mice (74 females per treatment group) were randomly assigned to the different F0 treatment groups and exposed via drinking water to 50 nM TBT or 0.1% DMSO vehicle (both diluted in 0.5% carboxymethyl cellulose in water to maximize solubility), for 7 days prior to mating as we have described[8,13]. One male was housed with two vehicle or 50 nM TBT exposed F0 females per cage to breed during the dark cycle (6PM to 6AM). Vaginal plug appearance was defined as embryonic day (E) 0.5. Treatment was removed during mating, then resumed for F0 females after copulation plugs detected (and males removed) then maintained until pups were born (Supplementary Fig. 11). This TBT concentration was chosen based on our previous studies[8,12,13,31] and is fivefold lower than the established no observed adverse effect level (NOAEL)[32]. While chemicals were administered to the dams throughout pregnancy, sires were never exposed to the treatment. No statistically significant differences were observed in the number of pups or the sex ratio per litter among the different groups (Supplementary Fig. 12). It should be noted that F2 descendants were exposed to TBT as germ cells in the exposed F1 embryos. F3 descendants were not exposed to TBT.

From each generation, we randomly chose only one male and one female per litter for endpoint analysis and another one male and one or two females per litter for breeding to produce the next generation (Supplementary Fig. 12). Animals selected for breeding the next generation were randomly chosen from the entire group of animals which passed the litter size criterion (6–8 pups, ≥2 of each sex per litter) and after eliminating the smallest and largest remaining animals (outliers) at weaning. Since our experiments used individuals from *n* = 16 litters for each treatment group and outliers were not included in the pool

from which random animals were selected, there should not be any outlier effects. There were insufficient animals in the F1 generation to both breed the F2 generation and analyze phenotypes in the diet challenge, so we only bred the F1 animals. To randomize the breeding process as much as possible, we did not breed siblings and never bred females from the same litter with the same male. Control animals were bred to each other, and TBT-exposed animals were bred to each other. There was no effect of treatment on litter size or sex ratio (Supplementary Fig. 13) as in our prior experiments. Each pregnancy was derived from a single male as demonstrated by pedigree analysis (Supplementary Fig. 17). All crosses were virgin crosses which was intended to eliminate any confounding effects of multiparous vs. virgin dams.

## Diet challenge and body composition analysis

Animals from control and treatment groups were maintained on a standard diet (SD) (PicoLab 5053; 24.5% Kcal from protein, 13.1% Kcal from fat, and 62.3% Kcal from carbohydrates) from weaning onward. In diet challenge experiments, F2 (14 males and 14 females for each group) and F3 (15 males and 15 females for each group) were switched to higher fat diet (HFD) (PicoLab Rodent Chow 5058, 23.2% kcal from protein, 21.6% kcal from fat, and 55.2% kcal from carbohydrates) whereas control groups (F2: 15 males and 15 females; F3: 12 males and 12 females) were maintained on the SD (PicoLab Rodent Chow, 5053). Body weight and body composition were measured weekly for each animal using EchoMRI™ Whole Body Composition Analyzer, which provides lean, fat and water content information. Total water weight includes free water mainly from the bladder and water contained in lean tissue. Littermates of the animals chosen for HFD analysis were used for breeding the next generation in every case; therefore, the animals chosen to breed the F2 and F3 generations were never exposed to HFD challenge. The choice of different times for the HFD experiment was based on an attempt to shorten the length of the experiment. While we typically used 12 weeks (See ref. 13) for the diet challenge, we shortened this to 5 weeks for this experiment based on another study of ours (Chamorro-Garcia et al. 2018) that only examined phenotypes in F1[31]. F2 descendants started diet challenge at 5 weeks of age for 8 weeks when a significant fat content increase was confirmed and persisted. F3 descendants starting the diet challenge at week 5 had not become significantly fatter by week 17. Therefore, we switched a group of reserved siblings of F3 animals to the HFD at week 17 for 5 weeks. Fat content was significantly increased in this group by week 19. Mice were fasted for 12 h prior to euthanasia and tissue collection.

Blood was collected via the saphenous vein at week 4 and week 12 (before and after diet challenge) for F2, and at weeks 4, 12, and 22 for F3. Blood was collected into heparinized tubes, then centrifuged for 15 min at 2000×$g$ at 4 °C. The resulting plasma was transferred to a clean tube and preserved at −80 °C. Animals were euthanized by isoflurane exposure followed by cardiac exsanguination after 4 h fasting. Blood was drawn into a heparinized syringe and centrifuged for 15 min at 2000×$g$ at 4 °C. The resulting plasma was transferred to a clean tube and preserved at −80 °C. We measured plasma leptin levels to confirm the previously reported phenotypes[8]. Inguinal white adipose tissue (iWAT), gWAT, pancreas, spleen, liver, interscapular brown adipose tissue (iBAT), and soleus muscle were flash frozen in liquid $N_2$ then stored at −80 °C for subsequent analysis. Feces were freshly collected from animals prior to, and during the diet challenge at week 4 and week 12 for F2, week 4, 12, and 22 for F3, and stored at −80 °C.

## PGC isolation

A randomly-selected subset of pregnant females was euthanized 13 days (E13.5) after vaginal plug detection. E13.5 embryos were isolated from euthanized pregnant dams, and E13.5 gonads containing primordial germ cells (PGCs) were isolated using a Leica MZ9.5 Binocular Stereo Microscope. Gonads were identified and sexed by their characteristic morphology at E13.5 (Supplementary Fig. 14a, b), and sex was verified by PCR[33]. Primer sequences are given in Supplementary Table 1. Gonads from same-gender embryos in each litter were pooled prior to tissue dissociation. Gonads were enzymatically digested using the Embryoid Body Dissociation Kit (Miltenyi Biotec). Next, total dissociated gonad cells were stained with Zombie Red Dye (BioLegend), PE Mouse anti-SSEA-1 (BD Pharmingen), and Alexa-Fluor 647 Mouse anti-CD61 (BD Pharmingen). Primordial germ cells were purified based on the expression of Zombie Red⁻/SSEA-1⁺/CD61⁺ using BD FACS Aria II Cell Sorter (BD Bioscience). Somatic gonad cells were purified based on the expression of Zombie Red⁻/SSEA-1⁻/CD61. The gating strategy and purity of the isolated cells is shown in Supplementary Fig. 15.

## Hi-C data generation

Five litters of each group with ~20,000 PGCs (Zombi Red⁻/SSEA-1⁺/CD61⁺) were designated to proceed for Hi-C sample preparation using Arima Hi-C Kit following the manufacturer's instructions for low-input Hi-C sequencing. This low-input protocol supported quantitative determination of topologically associating domains (TADs) from 10,000 human cells (Arima Genomics Application Note: "Unlock Low-Input 3D Genome Analysis with the Arima-Hi-C Kit", Arima Genomics), which was confirmed by our current study (Supplementary Fig. 2a–d). Briefly, cells were fixed with formaldehyde (Ted Pella) to crosslink chromatin contacts. Genomic DNA was isolated from the fixed cells and digested using a restriction enzyme cocktail. The digested 5′-overhanging ends were filled with biotinylated nucleotides, and spatially proximal digested ends were ligated. Proximally ligated DNA fragments, which capture chromatin contacts, were purified, fragmented by sonication, and enriched using streptavidin-conjugated beads. Illumina sequencing libraries were synthesized from the solid phase-captured DNA fragments using the Swift Biosciences® Accel-NGS® 2S Plus DNA Library Kit (Swift). Libraries were sequenced using an Illumina NovaSeq 6000 deep sequencer to obtain 150 + 150 nt paired-end FASTQ reads. Prior to Hi-C analysis of PGCs, we verified by PCR that each pool was exclusively comprised of male or female embryos. Supplementary Fig. 16 shows a table of X and Y chromosome counts from the Hi-C reads. Y chromosome counts for males were well balanced among all samples, intra- or intra-condition categories. Female samples showed very small but non-zero Y chromosome counts due to known strong similarities in the genomic DNA sequences between the X and Y chromosomes; these counts would not affect the interpretation of the Hi-C data. The X chromosome counts were approximately twice greater in females than males, which is expected, and well-balanced intra- and inter-conditions. Therefore, the numbers of Hi-C reads mapped to sex chromosomes were consistent with expectations.

## Identification of differential topologically associating domains (dTADs) and genes

After adapter sequences and low-quality reads (<30) were removed using the Trim Galore! tool, FASTQ reads were subjected to Hi-C seq analysis using the Hi-C Pro tool[34]. The Hi-C Pro quality control plots showed that at least 300 million valid interaction pairs were generated for each group of embryos (>60 million valid pairs per individual embryo) with greater than 60% long (>20 kb) cis interactions, indicating successful generation of sufficient amounts of high-quality Hi-C data (Supplementary Fig. 2e–h). Using these data, we were able to reproduce a known TAD profile surrounding the HoxD gene cluster[15], which was demonstrated using a contact map (Supplementary Fig. 2i) and a directionality index plot (Supplementary Fig. 2j). Locations of chromatin contact boundaries and the differential chromatin interaction (DCI) scores between TBT versus DMSO vehicle groups were calculated using the BART-3D software tool[35]. Distributions of TADs and dTADs were visualized using the OmicCircos[36] and the HiTC[37] R/

Bioconductor tools. For visualization, DCI scores in chromosome 19 were smoothened with 40 kbp bins and rolling averaged with 11 bins.

## Chromatin immunoprecipitation-quantitative polymerase chain reaction

Chromatin immunoprecipitation (ChIP) was performed using the method by Abcam and optimized with the established method we previously described[38]. Briefly, 50 mg of liver tissue that had been snap-frozen in liquid $N_2$ were thawed on ice in cold PBS and dispersed into single cell suspensions using a 100-µm cell strainer (#22363549; Fisher Brand, PA). Cells were washed twice with PBS containing protease inhibitor cocktail (#ab201111; Abcam, Cambridge, UK), then resuspended and fixed at room temperature for 10 minutes with 1% paraformaldehyde (Fisher Chemical, PA) in DMEM, followed by an ice-cold phosphate-buffered saline wash, and then quenched for 5 min with 125 mM glycine at room temperature. Fixed cells were washed, collected by centrifugation, then resuspended in phosphate-buffered saline at $10^7$ cells/mL. To isolate nuclei, cell pellets were lysed at 4 °C for 10 min with a gentle detergent recipe consisting of 50 mM HEPES-KOH, pH 7.5, 140 mM NaCl, 1 mM EDTA, 10% glycerol, 0.5% Nonidet P-40, 0.25% Triton X-100, Protease Inhibitor Cocktail (#ab201111; Abcam, Cambridge, UK). Nuclei were recovered by centrifugation at 8000×$g$ for 15 min, washed for 10 min at room temperature (10 mM Tris-HCl, pH 8.0, 200 mM NaCl, 1 mM EDTA, 0.5 mM EGTA, protease inhibitors (#ab201111; Abcam, Cambridge, UK), and lysed in 300 µL nuclear lysis buffer (10 mM Tris-HCl, pH 8.0, 200 mM NaCl, 1 mM EDTA, 0.5 mM EGTA, 0.1% Na-deoxycholate, 0.5% N-lauroylsarcosine, protease inhibitors (#ab201111; Abcam, Cambridge, UK). Chromatin samples were prepared by sonicating in 0.5 mL thin-walled polymerase chain reaction tubes (BrandTech, CT) using a QSonica Q800R2 (QSonica, CT) with the following settings: 30 s on/30 s off, amplitude 40% repeated for 30 minutes. Triton X-100 (1%) was added to sonicated lysates prior to high-speed, cold centrifugation to remove debris. A total of 5 µg DNA was immunoprecipitated with preblocked protein A/G Dynabeads (ThermoFisher Scientific, MA) complexed to 2.5 µg antibody (anti-CTCF, ab128873, anti-RAD21, ab217678, or Isotype IgG control, ab171870, Abcam, Cambridge, UK). Beads were washed three times with LiCl buffer (50 mM HEPES-KOH, pH 7.5, 500 mM LiCl, 1 mM EDTA, 1% Nonidet P-40, 0.7% Na-deoxycholate) and once with Tris-EDTA buffer plus 50 mM NaCl. To release chromatin from beads, pelleted beads were resuspended in elution buffer (50 mM Tris-HCl, pH 8.0, 10 mM EDTA, and 1% sodium dodecyl sulfate) and incubated at 65 °C for 30 min. Cross-link reversal was performed overnight at 65 °C. DNA samples were purified using Qiaquick PCR Cleanup kit (#28106, Qiagen, Germantown, MD) following RNase A (0.2 mg/mL, 2 h, 37 °C) and proteinase K (0.2 mg/mL, 2 h, 55 °C) treatment. Input DNA content was determined by spectrophotometry (Nanodrop, Thermo Fisher Scientific, MA). For analysis of candidate loci, real-time PCR was performed using SYBR™ Green PCR Master Mix (Thermo Fisher Scientific, MA) on a Roche LightCycler 480 II (Roche, Switzerland) according to the recommended protocol. Enrichment of the ChIP target was presented as a fold difference between specific Ab-immunoprecipitated samples and the immunoprecipitated total input with an IgG control. Primer sequences of the examined loci are listed in Supplementary Table 1. Multiple primer sets were tested for each site. For sites B and C, 2 of 2 primer sets showed significant enrichment. For binding site F, 1 of 3 primer sets showed enrichment and for site G, 2 of 3 showed enrichment.

## Quantitative real-time reverse transcriptase polymerase chain reaction (qPCR)

Tissue that had been previously snap-frozen in liquid $N_2$ was cut into ~20 mg pieces and lysed with Trizol following the manufacturer's recommended protocol (Thermo Fisher Scientific, MA); total RNA was recovered after isopropanol precipitation (Fisher Chemical, PA).

Complementary DNA was synthesized from 5 µg total RNA using SuperScript IV First-Strand Synthesis System (Thermo Fisher Scientific, MA) according to the manufacturer's instructions. Gene expression was assessed with real-time quantitative polymerase chain reaction (qPCR) using SYBR™ Green PCR Master Mix (Thermo Fisher Scientific, MA) on a Roche LightCycler 480 II (Roche, Switzerland). Primer sequences of the examined genes were listed in Supplementary Table 1. Cycle threshold values were quantified as the second derivative maximum using LightCycler software (Roche, Switzerland). The $2^{-\Delta\Delta Ct}$ method[39] was used to analyze RT-qPCR data and determine relative quantification corrected for primer efficiency. *Ide* expression was normalized to the housekeeping gene, *GAPDH*, and compared to DMSO descendants' group. Error bars represent the SEM from 15 to 17 biological replicates, calculated using standard propagation of error.

## Measurement of insulin and C-peptide

C-peptide serum levels were measured by EIA (Crystal Chem #80954; Elk Grove Village, IL, USA) at two different time points (before and after diet challenge) in plasma from blood samples drawn after overnight (12 h) fasting. Insulin levels were measured by EIA (Crystal Chem #90080; Elk Grove Village, IL, USA) in plasma from blood samples drawn after overnight (12 h) fasting.

## ChIP-seq analysis

ChIP-seq deep sequencing libraries were prepared from 10 ng purified genomic DNA fragments co-precipitated with chromatin enriched for CTCF (ab128873, Abcam Cambridge, UK and ChIP-IT Express 53008, Active Motif, Carlsbad, CA USA) using the NEBNext Ultra II DNA Library Prep Kit (New England Biolabs, Cat# E7103S) and sequenced with Illumina NovaSeq X Plus to generate 150 + 150 paired-end reads. FASTQ reads were subjected to quality assessment, adapter sequence removal, and filtering (Phred score >20) using FastQC. The filtered FASTQ files were aligned to the GRCm38/mm10 mouse reference genome using STAR[40] with spliced alignment disabled by "--alignIntronMax 1 –alignEndsType EndToEnd" options as we previously described[41]. The resulting BAM files were sorted and subjected to the extraction of uniquely mapped reads using SAMBAMBA[42]. From the deduplicated BAM files, normalized bigWig files were generated using deepTools[43] (bamCoverage) with binSize = 10, ignoreForNormalization = chrM, ChrX, for the mm10 genome size. The bigWig data were visualized using the Integrative Genomics Viewer[44] for direct assessment of CTCF ChIP-seq profiles of chromosome 19. Peaks corresponding to the repetitive sequences were identified using RepeatMasker[45] and excluded from the analysis.

## ddPCR CNV analysis

Genomic DNA was isolated from snap-frozen liver tissue of F2 and F3 offspring using the Quick-DNA Miniprep Plus Kit (Zymo Research, D3025). DNA quality was confirmed by NanoDrop 2000 UV/Vis Spectrophotometer (Thermo Scientific), and 200 ng total gDNA per sample was sent to the Van Andel Institute Genomics Core (Grand Rapids, MI) for droplet digital PCR (ddPCR) analysis. Copy number of the *Ide* locus was measured using primers within the *Ide* gene and normalized to *Kif11*, which lies outside the reported CNV region. Raw CNV values returned by the core were analyzed by one-way ANOVA to test for group differences, and copy number values were plotted against *Ide* expression levels measured in the same samples. Results are shown in Supplementary Figs. 22 and 23.

## Ethics Statement

All experiments were conducted in compliance with all relevant ethical regulations. Animal procedures conducted in this study were approved by the Institutional Animal Care and Use Committee of the University of California, Irvine. Experiments utilizing chemicals,

biohazardous materials and recombinant DNA were approved by the Institutional Biosafety Committees at UCI and MGB.

## Reporting summary

Further information on research design is available in the Nature Portfolio Reporting Summary linked to this article.

## Data availability

The Hi-C and ChIP-seq data generated in this study have been deposited in the NCBI Gene Expression Omnibus (GEO) under accession codes GSE218701 and GSE268954, which are publicly available. Processed data supporting the findings of this study are included in the Supplementary Information. Source data are provided with this paper.

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

## Acknowledgements

Correspondence and requests for materials should be addressed to B.B. (blumberg@uci.edu) or T.S. (shioda@mgh.harvard.edu). Supported by grants from the National Institutes of Health, USA (R01ES023316 and R01ES031139) and the John Templeton Foundation to B.B. and T.S. We thank Drs. Sha Sun (University of California, Irvine, CA USA), Xiao-Min Ren (Kunming University of Science and Technology, China), Angel Nadal (Universidad Miguel Hernández de Elche. Spain), Ivan Quesada (Universidad Miguel Hernández de Elche. Spain), Hanjun Lee (Whitehead Institute, Cambridge, MA, USA), and Rob Lustig (University of California, San Francisco, CA USA) for critically reading the manuscript and Dr. Hanjun Lee for suggesting BART-3D. We wish to thank Drs. Raquel Chamorro-Garcia (University of California, Santa Cruz) and Grant MacGregor (University of California, Irvine) for advice and assistance during the early stages of this project. We are indebted to Dr. Andrew Pospisilik (Van Andel Institute), who informed us of the CNV issue and had the ddPCR analysis of copy number in the Ide region performed at the Van Andel Institute genomics core under his funding.

## Author contributions

R.C.C., R.J.E., T.S., and B.B. designed the experiments reported in this manuscript. R.C.C., R.J.E., A.A.M., Z.L., A.L.D., K.T.T., M.A., E.M.J., H.B.W., A.N., K.S., J.O., T.S., and B.B. performed experiments. R.C.C., M.S.L., T.S. and B.B. analyzed data. R.C.C., T.S., M.S.L., and B.B. wrote the manuscript.

## Competing interests

B.B. is a named inventor on US patents related to PPARγ. The remaining authors declare no competing interests.
