## [Transparent Peer Review file · Nature Communications]

Heritable Changes in Chromatin Contacts Associated with Transgenerational Susceptibility to Diet-Induced Insulin Dysregulation and Obesity

Corresponding Author: Dr Bruce Blumberg

Version 0:

Reviewer comments:

Reviewer #1

(Remarks to the Author)

In this manuscript, by using a mouse model, the authors reveal that exposure to the obesogen tributyltin (TBT) induces heritable chromatin interaction changes in primordial germ cells (PGCs). New interactions within the *Ide* gene are formed in directly exposed PGCs, persisting in subsequent unexposed generations. This leads to decreased *Ide* mRNA expression in descendants' livers, resulting in male-specific hyperinsulinemia, hyperglycemia, and increased fat mass. These findings propose a molecular basis for transgenerational obesity predisposition due to gestational exposure to an environmental obesogen, offering insights for future studies on chromatin structure alterations across generations in mammals. The difference between control and experimental groups are significant and figures are well organized. I listed some comments below for the authors to consider before publication.

1. In the authors' prior research, insulin levels remained unchanged in F4 descendants of DMSO- and TBT-treated animals at 33 weeks of age. However, the authors observed significantly higher plasma insulin levels in TBT group males compared to the DMSO control. Could the variance be attributed to the different ages of mice used in the experiments?
2. Consider relocating Extended Data Figures 8 and 9 to the main figures, as they provide crucial insights into the increased insulin levels observed in TBT-group males both before and after high-fat diet (HFD), suggesting a potential link to impaired insulin clearance.
3. The background in the introduction section should be more comprehensive, given the recently rapid growth of intergenerational/transgenerational inheritance reported in mammals.
4. This authors are suggest to discuss the current finding into context, for example, recent reports using a model of BPA-induced obesity also found multi-generational inheritance in mice, similarly reveals the contribution of heritable long-range chromatin interactions, which is synergized with altered DNA methylation and RNA/RNA modifications (PNAS 2022 PMID: 36469784; Nat Rev Endocrinol 2023 PMID: 36792712)

Reviewer #2

(Remarks to the Author)

- 1) The topic of transgenerational epigenetic inheritance and molecular mechanisms involved is an interesting topic and role of chromatin structure is a critical mechanism to consider. The observations provided and the data presentation and experimental design are novel and appropriate for consideration of publication. This is one of the first studies to demonstrate the impact of chromatin structure alterations in epigenetic transgenerational inheritance, so is also appropriate for publication.
- 2) The data presented, analyses, and statistical considerations were also appropriate and well described in the Methods and Results sections. No suggested modifications are required in the Results or Methods sections.
- 3) The primary issue with the manuscript is the complete bias of the authors in the discussion and perspective provided for the manuscript. The authors state in lines 22-25, 36-38 of the Abstract, 46-49 Introduction, and 235-242 Discussion that no

previous studies provide mechanistic information on the phenomenon of epigenetic transgenerational inheritance. Over 5,000 publications on the topic exist and over 2,000 specific publications address the mechanistic aspects of transgenerational inheritance from drosophila to humans. The combined actions of different epigenetic facts such as DNA methylation, histone modifications and ncRNA in this process has been demonstrated. However, the author states these studies are not relevant and the current study somehow reveals the mechanistic aspect of transgenerational inheritance in the text referenced above. This is not accurate and intentionally disregards the extensive literature on the topic. Although this language may have prompted the editors to review the paper, in its current form is extremely misleading. Such a misrepresentation to the previous literature and the true context of the data presented requires revision to consider for publication.

4) The term “chromatin contacts” or “chromatin interactions” (CI) is also poor vocabulary and misleading. Out of over 100,000 manuscripts the reference to “chromatin structure” is used to reflect the 3D loops or TADs in reference to chromatin, while the terms chromatin interactions and contacts are reflected in the over 20,000 studies to binding of ncRNA, histone modifications prompting protein interactions, or DNA methylation promoting various chromatin remodeling proteins to allow the chromatin structure to alter. Therefore, the terminology used for chromatin interactions (CI), or contacts are misleading and should be “chromatin structure” in reference to TADs as presented. This vocabulary needs to be corrected throughout the manuscript to put the paper in a better context and less misleading interpretations. Clearly no epigenetic element, including chromatin structure acts independently, such that DNA methylation, ncRNA and histone modifications are required to get any defined chromatin structure. This needs to be thoroughly revised throughout the manuscript.

5) Although the manuscript requires a better vocabulary, context, and inclusion of all epigenetic factors to be accurate and non-biased, this reviewer feels the manuscript can be thoroughly revised to present the data obtained and put into a thoughtful and more accurate context.

Reviewer #3

(Remarks to the Author)

Chang and colleagues report in this manuscript that exposure to tributyltin (TBT) causes heritable changes in chromatin interactions (CIs) in PGCs in F1s, which can be stably maintained in PGCs and the liver of the subsequent unexposed F3 generation. In the only DCI identified to be persistently maintained across three generations, the downregulation of *Ide* in the liver was shown to correlate with the hyperglycemic and hyperinsulinemic phenotype in F3 mice. The authors concluded that CI changes may represent a molecular mechanism underlying the transmission of the transgenerational predisposition to obesity due to gestational exposure to an environmental obesogen. While the idea that changes in chromatin configuration underlie transgenerational epigenetic inheritance is of great interest, it is not completely novel, and the data shown in this report are largely correlative.

General:

This study raises more questions than answers. For example, why was only one DCI maintained in F3 while 20 ICs were detected in F1 and F2? Why are only males affected? Why were the DCIs detected in PGCs maintained in the liver but not in other organs? Are other genes in the only DCI dysregulated? If so, what are the contributions of those genes? Are those DCIs detected in PGCs maintained in sperm? This reviewer understands that these may be topics for future studies. The large gaps between DCIs and the observed phenotype leave the study purely correlative and hard to believe.

Specifics:

Validation of CI formation in the *Ide* gene was conducted using ChIP-qPCR, which can be biased. Wouldn't it be better to use ChIP-seq? ChIP-seq would allow for the validation of the Hi-C data as well.

The title and the text all mentioned obesity as the phenotype, but the data showed no significant changes in either body weight or composition. The changes detected reflect glucose metabolic disorders.

Given that significant reprogramming occurs during the meiotic phase of spermatogenesis (NCB, 2023, 25:1520), some of the CI changes may be reset and thus disappear in sperm. Are those detected DCIs present in sperm as well?

Reviewer #4

(Remarks to the Author)

This manuscript describes experiments to examine whether 3-D genome interactions as detected by HiC could potentially be involved, or associated with multigenerational phenotypes observed in offspring of mice treated with TBT. Overall the manuscript is very clearly written considering the challenge often associated with communicating inter/transgenerational literature. The phenotyping data appear well done. Overall the data are correlative without a test of causality or replication. The phenotypes themselves raise no concerns and are in keeping with obesity-prone developmental programming, and therefore are important. The genomic data and experimental design raise a number of key questions that leave the impact of the work unclear. These issues are of high importance given that the entire field currently suffers from the legacy of such critiques. It is therefore important at this level of publication that the experiments, data and design rise up to a definitive level. The authors are lauded for the work invested. My significant concerns in order they arose:

- Can the authors please show a pedigree of the experiment including all animals and color code perhaps which animals

were used for what experiments? It is unclear why so many mice were originally ordered and what the number of biological replicates is for the Hi-C. Is it an n=1 pooled from many? Paternal variation appears not to have been eliminated by having an equal number of DMSO and TBT pregnancies derived from each male. Is this true? The fact that paternal intergenerational effects exist and that they have been suggested to potentially elicit CTCF-dependent, should be mentioned. Similarly, given that virgin/primiparous mothers lactate and behave differently than multi-parous mothers, were these all virgin crosses?

- How are animals chosen? Are they the most median-like for their respective group (eg. the most average sex-specific body weight for the litter? Randomized sampling, when overall sample number is low, 'needlessly' includes outliers at a frequency that 'randomly' may impact one group more than another. This chance is reduced as sample number increases. Similarly, it provides a robust measure of control to experiments for a field that is often criticized for such design (eg. inter-/transgenerational effects).

- What is the rationale for the different times of HFD treatment in F2 and F3. Given the early timepoint associated with HFD in F2, does this mean the F3 are derived from F2 that also experienced HFD?

- It sounds like the animals in Fig 1b were also exposed to HFD at 5w of age but this is not indicated on the graph. HFD is well known to have substantially different effects when applied early vs late, and HFD creates substantial metabolic effects irrespective of weight-gain.

- The metrics used to conclude that the Hi-C profiles are globally similar with exception of these detected DCI's are insufficiently self-critiquing. There is no evidence of intra-condition variability against which to gauge inter-condition variability.

- X and Y chromosome mapping counts across samples should be shown given the pooling approach.

- Embryonic sufficiency and phenotypes co-vary with the position of each embryo relative to blood supply, with those animals closest to the arterial supply being larger than those more distal. In animals that undergo natural birth, this information is lost and inaccessible. In animals that are taken at fetal stages however this variable can be controlled for. Was embryonic position accounted for?

- One alternate explanation for constellation of results specifically at the *Ide* locus is a genomic structural rearrangement. Some translocations/inversions, and in particular genomic duplications, can trigger the same pattern of HiC results and longevity of phenotypic changes. This reviewer is not an expert in SDquest; is the tool robust enough to rule out that a subset of animals within a pooled replicate contain an SV? Even a minor fraction of such events should yield a robust and reproducible signal in HiC (and phenotypes). This could be approached by examining even two independent biological replicates from distinct families/pedigrees (assuming this is not a striking hotspot event). Again, are there replicates (from distinct pedigrees) that could help rule this out? Alternatively, if the authors still have DNA from other tissues of the specific F2 and F3 animals involved, a methylation array analysis, WES, DNA microarrays could all be used to assess copy number across individual animals to rule this in or out presumably without enormous costs (or even in technical replicates of the pooled samples).

- Do the authors discount potential roles for the many other DCI's observed that didn't survive the approach?

- Given the issues above, the fairly strong implications about causality should be toned down (or eliminated until causality is tested) and replaced with alternate explanations / discussions of mechanisms through which these changes would be expected to be repeatedly induced and how they might survive (or be saved/protected) through epigenome erasure and development. Do the authors suggest that these changes are maintained through toti- / pluripotency? Do they have corroborative evidence for instance using FISH that these contacts are indeed increased?

- Early post-weaning period is a phase of life still considered development and overlapping with substantial endocrine fluctuations, final maturation of the gonads, as well as puberty/menarchy. HFD appears to have been administered during this time period in one set animals (F2, if I understand the methods correctly). HFD is well known to impact 5-8 w old mice very differently than adult mice. Also, mice immediately post-weaning have just moved from mixed milk/chow to chow-only diet and their metabolic tissues are undergoing a related wave of environmentally triggered terminal maturation and their microbiome is also highly variable between animals. Was there a rationale I am missing to start an intervention given the high degree of unmeasured variability and developmental rearrangement that is happening in this time window? With all due respect, this is truly unfortunate for an experiment focused on variation/plasticity. All figures/text should make clear that F2 underwent 2 interventions. The authors are applauded for the transparency in the report.

Minor

- No DMSO/TBT legend on Ext data 10, unless I missed it.

Version 1:

Reviewer comments:

Reviewer #1

(Remarks to the Author)

The authors have addressed all my questions.

Reviewer #2

(Remarks to the Author)

The revisions did not address my concerns. The texts, like sentence 2 and 3 of abstracts, state epigenetic transgenerational inheritance is questioned and no mechanistic data has been provided. This is incorrect after over a 1000 publications on the topic with all epigenetic components known shown to be involved including chromatin structure. So the

manuscript needs to be put in perspective with the current literature and not the bias of the investigators. Remove this type of text throughout the manuscript and put it in perspective to the actual extensive literature on the topic. To suggest this is the first study to provide a mechanism on the phenomenon and no other mechanisms have been shown to be involved is not accurate and needs to be revised. A number of studies do already show chromatin modifications and structure are regulated by other epigenetic factors and they all work together to promote transgenerational inheritance. So please provide a revision that has a more appropriate perspective and considers the literature on the topic. The data provided are very good, but the manuscript needs to be revised.

Reviewer #3

(Remarks to the Author)

The authors have addressed most of my concerns, and this review has no further questions.

Reviewer #4

(Remarks to the Author)

Comments to the authors:

I must apologize for my delay in submitting this response. I had too many competing obligations so timing was poor. I appreciate the patience and apologize to the authors and editor. The manuscript remains one of overall high quality, with potentially interesting results. Powerful aspects of the work include the very systematic evaluation of changes in PGCs as well as a regular somatic tissue and the authors are commended for their work. My previous concerns are partly addressed. Claims of causality relating specifically to *Ide* remain untested. As written, the potential effects of other loci, and of the other genes within and just outside the regulated TAD are ignored and this does a disservice to the authors and to the readers. These points are expanded in somewhat more detail below:

1. The authors have addressed my questions on experimental design and in particular my questions about breeding schemes, mouse selection, etc. Often their answers to my questions in the rebuttal are clearer than the manuscript methods and results so I encourage them to spell out some of the concerns with an almost verbatim copy and paste from their responses.
2. Regarding Hi-C interpretations. The individual replicate data should be shown, ideally with matched experimental partners. This is the only way to allow the reader to appreciate consistency of the response across pools. And this is important, as the authors must acknowledge that it will strike many readers as too good to be true that one and only one differentially regulated region was found; the over-argumentation for a very simple *Ide*-dependent causality do not help their cause in that context.
3. On that note of causality (also brought up by other reviewers), the fact that *Ide* 'fits' the narrative well does not rule out correlation or consequence. The authors seem to argue that the fact that they find the same DCI pulled out several times supports causality, and that it somehow argues against other explanations driving the phenotypes. I don't understand the logic. Their approach looked for reproducible changes so it can find reproducible changes. The equally likely explanation (other than a false positive which would require a repeat of the entire experiment to rule out – too much) for the locus change is that it is the result of the transgenerational metabolic phenotype induced by TBT. I.e. The changes could be the result of the strong glucoregulatory phenotype or any other non-measured phenotype. A recently funded R01 is irrelevant; it does not argue for or against *Ide*'s direct causal involvement. A quick search of *Ide* in GEO Profiles suggests it changes or is flagged as potentially relevant to >10,000 deposited datasets and the vast majority of these (by quick browsing) have little to do with the phenotypes or interventions discussed here. Therefore *Ide* also changes in response to a very wide range of interventions, data that support the possibility that *Ide* differences can be secondary. Even if the authors repeated everything again, and saw all the same things, it would still not change that. It would simply mean that the correlation (cause or consequence) is reproducible. The goals to use CRISPRi/a are noble and the data generated will be interesting. It will be important to manipulate those other genes too in this reviewer's view. For the purposes of this manuscript, this is simply an exercise of reducing the excessive focus on *Ide* (see below) and rewording.
4. Along the same lines, it seems inappropriate to treat the much larger changes at *Kif11* and *Hhex* (and possibly at other neighboring genes within the TAD) as unlikely mediators. *Kif11* deletion is embryonic lethal and mutation in humans triggers multi-system developmental disorder so it carries substantial phenotypic potential. *Hhex* deletion in mice triggers many phenotypes and is necessary for liver development (by conditional deletion) for hepatoblast differentiation and for bile duct morphogenesis. It would be surprising if those changes did NOT influence liver function (including insulin clearance). Similarly, why are there no measures of potential gene expression changes at *Fgf3*, *Btaf1*, *Cpeb3*, and *March5*? An RNAseq experiment of the extent of transcriptional rewiring that exists in these tissues, and the relative effect size of these changes to others would be of substantial value. Perhaps the authors have such data? The genes immediately outside (up and downstream of the TAD) are also not measured though they can / would be expected to be equally affected. Indeed, it is most likely that the contact rearrangement is inducing silencing on one direction (*Ide* direction) and activation in the other (eg. the *Hhex* *Kif11* direction). And all of these may be equally causal to the multiple evidenced phenotypes. *March5* heterozygotes shows evidence of bodyweight phenotypes with an apparent sex effect in males for instance based on data base searches. At a minimum, qPCR of the set of genes above, and the flanking genes should be performed in F2/3 PGCs and liver, should sufficiently inform the reader of the extent of the transcriptional changes evoked at this locus.
5. Reviewer 3 commented that the phenotypes reported are not really obesity, but rather glucoregulatory. I agree completely

with this statement. There is very little evidence of outright obesity in the model, rather a **susceptibility to** DIO that is relatively mild. This is an important and much more accurate nuance. The glucoregulatory phenotype is much stronger and doesn't require induction. That issue does raise whether this is not a transgenerational phenotype (F2/F3) but rather that TBT triggers a repeating F1 intergenerational phenotype and this should be discussed. Having the emphasis on obesity is somewhat misleading given the much stronger glucoregulatory changes. The title is currently incorrect.

Version 2:

Reviewer comments:

Reviewer #2

(Remarks to the Author)

The previous revisions have improved the manuscript and I appreciate the comments provided in the revision comments section. Although I feel the manuscript should be published, the following comments are provided to allow a further revision to improve the manuscript and put it into more accurate and less biased context. I feel this would improve the manuscript and provide a less biased presentation for the reader, but this is simply a suggestion for the author, and should not prevent publication.

1) Abstract, the controversial comment in the Abstract is accurate, but this is not due to the extensive literature on the topic of transgenerational inheritance, but this is primarily due to the concept of genetic determinism, which is predominant in today's science. Therefore, a discussion of controversy without mention of this issue is misleading.

2) New literature on the lack of DNA methylation erasure is not presented in regards to the comment DNA methylation is erased during embryogenesis. Observations now indicate the lower density DNA methylation that constitutes the majority of DNA methylation is not erased in the primordial germ cell, as occur with high CpG density sites. The transgenerational literature on ncRNA is also not referenced. The reviews listed are primarily older, so inclusion of new literature would be useful, lines 50-64.

3) Stable generational changes for DNA methylation and ncRNA have been shown in the literature, but this literature is ignored and states TADs only evidence, lines 256-264. Literature clearly demonstrates corresponding DNA methylation and histone modifications are required for chromatin structure and TADs to form, but this is not clarified in the Abstract or Discussion sections.

4) The bias to TADs or chromatin structure needs to be reduced. The clarification all epigenetic mechanisms are integrated and will be involved in epigenetic transgenerational inheritance needs to be clarified. The literature on this topic is not presented.

Reviewer #4

(Remarks to the Author)

This reviewer appreciates the author's response and efforts. They have addressed most of the concerns. There some remaining issues below, and with apologies, one issue of high relevance that must be brought up again.

Revisiting copy number variation (CNV)

Yesterday when presenting my own developmental plasticity work, a postdoc in the crowd asked if I was aware of the following paper by Watkins-Chow and Pavan - <https://genome.cshlp.org/content/18/1/60.full>. I was not. In reading it today on the flight I see it shows what appear to be convincing evidence of a dual signal CNV in Jax C57BL6J animals. The dual CNV signal precisely overlaps the signal identified in the manuscript under review here (ie. two likely-coupled CNV signals that match exactly the two Hi-C signals at *Ide* and *Fgf3*, one shorter one over *Fgf3*, and the second, slightly longer one, directly over *Ide*. Watkins/Pavan describe the CNV as being present at frequencies consistent with the possibility that they underpin the transgenerational effects described here. Presumably the origin of the problem was that Jax maintains extremely large colonies that make maintaining 'isogenicity' near impossible when generating and maintaining their colonies. This reviewer is unaware of what Jax has done since to remedy that situation. We will also be reaching out to Jax for clarity. On quick survey, I cannot find evidence that they have remedied the situation, and the postdoc who informed me of this indicated that they personally detected the CNV in many mice, ordered from Jax in 2020. These data raise significant concerns of a genetic underpinning for the current manuscript that have to do with the isogenicity/fidelity of C57BL6J line itself.

Because of the near impossibility of coincidence, particularly the exact overlap of both CNV signal-elements with the Hi-C signals identified here in this work, this begs a serious re-examination of the possibility that CNV variation underpins the incomplete penetrance. This reviewer would recommend replicating some or all of the techniques used by Watkins/Pavan to test the issue. This must of course be done on samples from individual mice (not pools, and ideally those used in the study), and from tissues that do not contain large variation in ploidy that may mask the signal (ie. definitely not liver; ideally haploid sperm... I guess?... I am not a CNV expert, yet). The authors should also consider long-read sequencing as well as using a read mapping/alignment strategy that allows for chimeric- read mapping.

Regardless, given the essentially perfect overlap with the reported CNV signals, documenting these validation results and clearly relating them to how/when the C57BL6J colony used here was distinguished from Jax (or relative to what Jax has

done, if anything, about the issue) is necessary. As a scientist, I feel this is completely inappropriate for Jax not to highlight boldly in their ordering system if it is still happening. We will also begin carefully looking into this in our lab. I feel for the authors and in many ways hate to have to be the one that brings it up.

In the case that the above issue turns out not to be an issue, the following points remain:

From last review Point 4 that highlighted potential relevance of the numerous genes within the TAD:

The authors responded: "To address the reviewer's comment, we now include qPCR analysis of the expression of FGFbp3, Btaf1, Cpeb3 and March 5 (Extended data Figs. 19, 20). With the exception of a slight increase in hepatic expression of Cpeb3 in males (~1.3 fold) and a small decrease in expression of Btaf1 in gWAT of F3 males, there were no significant changes observed in the expression of the indicated genes. Therefore, the hypothesis that these genes are equally affected to Ide or equally causal to the multiple evidenced phenotypes is not supported."

The authors are commended for their work. Because it is beyond the scope of this work to rule out involvement of the TAD-embedded candidate genes across all relevant metabolic tissues and cell-types, something like the following idea should simply be added to the results and discussion: 'Our findings are consistent with a mechanism that involves altered hepatic and Cebp3. The data do not rule out involvement of other genes within the TAD in other metabolic tissues, for instance through central (eg. hypothalamic) or immune regulation. The Watkins/Pavan paper actually highlight the earlier point, brain and spleen appear to be relevant sites of altered expression in those CNV varying mice.

From last review Point 5:

The authors responded "In response to this comment, we amended the title to "Heritable changes in chromatin contacts linked to transgenerational susceptibility to diet-induced obesity". The major change observed is elevation of plasma insulin as a result of decreased insulin clearance, particularly in response to increased dietary fat. Therefore, we believe that the revised title is descriptive of what the manuscript presents."

The authors state themselves that the major change is hyperinsulinemia and reduced insulin clearance, so why not put that in the title ("susceptibility to diet-induced insulin dysregulation")?. Those are the most substantial phenotypes; they ARE relevant to metabolic syndrome; they ARE metabolic sequelae of HFD; and they are NOT obesity (which as indicated has a really minimal phenotype). This reviewer would still recommend a more objective/accurate title.

Version 3:

Reviewer comments:

Reviewer #2

(Remarks to the Author)

The revised manuscript is unfortunately not improved, and the specifics are listed below.

Question on existence and mechanism of epigenetic transgenerational inheritance.

Instead of addressing the concerns, the authors have added an increased number of inaccurate comments and speculations. This includes the following:

Line 21 – Burgeoning evidence suggestion

Lines 23-25 – Existence unclear TEI

Line 27 – Unanswered questions

Lines 28-29 – Attempts but not successful

Line 42 – All other mechanisms not plausible

Lines 49-50 – Possible mechanisms not valid

Lines 53-54 – No previous literature support mechanism TEI

Lines 62-64 – ncRNA role uncertain

With hundreds of manuscripts on each of these topics, the authors discount all previous science and research on epigenetic inheritance to suggest this study is the only one valid or important. Suggests interpretation all previous studies is wrong in broader perspective.

The data and discussion of data in Results section in general is fine and well presented.

The Discussion section also presents derogatory comments on past research on the topic.

Lines 268-269 – Past conclusions TEI not accurate

Lines 270-274 – Involving a variety of epigenetic mechanisms suggested not correct in past literature.

Line 272 – A single type of epigenetics involved not correct.

Therefore, the authors neither accept the past literature and for the most part disregard the past 25 years of research. For example, they disregard ncRNA mediated DNA methylation or DNA methylation mediated chromatin structure and the role of these processes in the formation of TADs, which is the focus of their study.

They also disregard the literature that all genetic mutations, like CNVs, have a precursor epigenetic element. For CNVs, the removal of the DNA methylation promotes the CNV expansion, or ability of transposable element to move, or role ncRNA in methylation, or role methylation in the genetic point mutations like C-T conversions. The manuscript suggests a lack of any past literature and integration of epigenetics and genetics and role of the different epigenetic components in genome biology, including chromatin structure.

Reviewer #4

(Remarks to the Author)

Version 4:

Reviewer comments:

Reviewer #4

(Remarks to the Author)

Well done to the authors. I commend them for their stamina and patience with the outstanding issues and for their professionalism in pursuing as clear as possible a view on the potential of genetic variation to be a confounder. I have no further comments other than recommending that they remove the superlatives (strongly, highly) from their new CNV associated text, and let the data speak for themselves.

Reviewer #1 (Remarks to the Author):

In this manuscript, by using a mouse model, the authors reveal that exposure to the obesogen tributyltin (TBT) induces heritable chromatin interaction changes in primordial germ cells (PGCs). New interactions within the *Ide* gene are formed in directly exposed PGCs, persisting in subsequent unexposed generations. This leads to decreased *Ide* mRNA expression in descendants' livers, resulting in male-specific hyperinsulinemia, hyperglycemia, and increased fat mass. These findings propose a molecular basis for transgenerational obesity predisposition due to gestational exposure to an environmental obesogen, offering insights for future studies on chromatin structure alterations across generations in mammals. The difference between control and experimental groups are significant and figures are well organized. I listed some comments below for the authors to consider before publication.

1. In the authors' prior research, insulin levels remained unchanged in F4 descendants of DMSO- and TBT-treated animals at 33 weeks of age. However, the authors observed significantly higher plasma insulin levels in TBT group males compared to the DMSO control. Could the variance be attributed to the different ages of mice used in the experiments?

We appreciate the reviewer's careful assessments of our prior and current studies. As the reviewer noted, hyperinsulinemia is frequently associated with aging, which may contribute to insulin resistance in the elderly. Marmentini et al (PMC8150109). meticulously compared insulin metabolism of male C57BL/6 mice – which is the same strain we used in our past and current studies – between 3 and 18 months of age and found aging-associated increase in plasma insulin and decrease in insulin clearance. Strikingly, their study identified a decrease in hepatic *Ide* expression and IDE enzymatic activity as a major contributor to the observed age-associated hyperinsulinemia. Therefore, in our prior study, it is likely that the IDE-dependent hepatic insulin clearance was diminished in both DMSO- and TBT-group male C57BL/6J mice at 33-weeks of age, thus obscuring the TBT effects on hepatic IDE expression. We also noticed that coprophagy could have also affected fasting plasma insulin level in our prior experiments, which has been addressed in our current study. As we continue efforts to optimize the animal experiment protocol, effects of F0 TBT-exposure on fasting plasma insulin levels in the male descendants is becoming evident. We added a brief discussion on this matter in the revision.

2. Consider relocating Extended Data Figures 8 and 9 to the main figures, as they provide crucial insights into the increased insulin levels observed in TBT-group males both before and after high-fat diet (HFD), suggesting a potential link to impaired insulin clearance.

We have incorporated male data from Extended data Figures 8 and 9 into Figure 4 at kept the female data as a new Extended data Fig. 10.

3. The background in the introduction section should be more comprehensive, given the recently rapid growth of intergenerational/transgenerational inheritance reported in mammals.

We have attempted to increase the comprehensiveness of the background although the space allowed for this part in the manuscript is limited due to the journal's formatting rules.

4. These authors are suggested to discuss the current finding into context, for example, recent reports using a model of BPA-induced obesity also found multi-generational inheritance in mice, similarly reveals the contribution of heritable long-range chromatin interactions, which is synergized with altered DNA methylation and RNA/RNA modifications (PNAS 2022 PMID: 36469784; Nat Rev Endocrinol 2023 PMID: 36792712)

We thank the reviewer for this suggestion and have incorporated both references into our discussion.

Reviewer #2 (Remarks to the Author):

1) The topic of transgenerational epigenetic inheritance and molecular mechanisms involved is an interesting topic and role of chromatin structure is a critical mechanism to consider. The observations provided and the data presentation and experimental design are novel and appropriate for consideration of publication. This is one of the first studies to demonstrate the impact of chromatin structure alterations in epigenetic transgenerational inheritance, so is also appropriate for publication.

We thank the reviewer for these positive comments.

2) The data presented, analyses, and statistical considerations were also appropriate and well described in the Methods and Results sections. No suggested modifications are required in the Results or Methods sections.

We also thank the reviewer for these supportive comments.

3) The primary issue with the manuscript is the complete bias of the authors in the discussion and perspective provided for the manuscript. The authors state in lines 22-25, 36-38 of the Abstract, 46-49 Introduction, and 235-242 Discussion that no previous studies provide mechanistic information on the phenomenon of epigenetic transgenerational inheritance. Over 5,000 publications on the topic exist and over 2,000 specific publications address the mechanistic aspects of transgenerational inheritance from drosophila to humans.

As the reviewer points out, many mechanisms have been discussed in the literature with varying strength of evidence in multiple organisms. However, with respect to environmentally caused non-genetic inheritance of disorders in mammals, there are very few studies that have convincingly demonstrated a heritable mechanism underlying the specific phenotypes. Whereas changes in DNA methylation or histone modifications may contribute to the transgenerational phenotypes, no differential epigenetic marks of these types have been identified at the same loci in the same tissues across generations. The lack of such demonstration has been a major problem in this field of research despite the large numbers of publications on this subject. Contrary to this review comment, we carefully attempted to present our assessment on the current status of research on mechanisms of transgenerational epigenetic inheritance (TEI) without introducing bias. To us, it sounds rather biased to state that the mechanisms of TEI have been established, or are even becoming evident, based on thousands of publications on this specific subject. Thus, to address this critique, we added the following to the introduction and qualified throughout the manuscript that we are referring to TEI in mammals:

“Thus, despite a large number of previously published studies proposing a wide variety of mechanistic insights into the phenomenon of epigenetic transgenerational inheritance (TEI), transgenerationally preserved changes in conventional epigenetic marks (such as DNA methylation or histone modifications) at specific loci are yet to be identified in mammals. The lack of this critical evidence has been a major obstacle to establish TEI as an alternative mode of inheritance in mammals.”

The combined actions of different epigenetic factors such as DNA methylation, histone modifications and ncRNA in this process has been demonstrated. However, the author states these studies are not relevant and the current study somehow reveals the mechanistic aspect of transgenerational inheritance in the text referenced above. This is not accurate and intentionally disregards the extensive literature on the topic. Although this language may have prompted the editors to review the paper, in its current form is extremely misleading. Such a misrepresentation to the previous literature and the true context of the data presented requires revision to consider for publication.

As noted above, combined action of different epigenetic phenomena has been suggested to underlie transgenerational inheritance. However, it remains to be demonstrated that the same marks, be they DNA methylation or histone modifications, are present in the same genes at the same locations from generation to generation. Rather the opposite has been shown – namely, whereas changes in DNA methylation, histone modification and ncRNA expression have been shown to exist from generation to generation, they are rarely, if ever the same marks at the same loci. The extensive studies of Skinner and colleagues showed the absence of the same marks across generations although, changes are observed in different places in different generations.

To address this critique, we rephrased our discussion to the following:

“Associations of several different epigenetic events with TEI have been proposed in preceding studies. However, whether such associations – either as independent events or in combination – are relevant to the mechanistic basis of TEI has not yet been established. To provide mechanistic supports for mammalian TEI, evidence is required that such epigenetic events are preserved across generations at specific loci and likely affect expression of genes that are directly relevant to the observed TEI phenotypes.”

4) The term “chromatin contacts” or “chromatin interactions” (CI) is also poor vocabulary and misleading. Out of over 100,000 manuscripts the reference to “chromatin structure” is used to reflect the 3D loops or TADs in reference to chromatin, while the terms chromatin interactions and contacts are reflected in the over 20,000 studies to binding of ncRNA, histone modifications prompting protein interactions, or DNA methylation promoting various chromatin remodeling proteins to allow the chromatin structure to alter. Therefore, the terminology used for chromatin interactions (CI), or contacts are misleading and should be “chromatin structure” in reference to TADs as presented. This vocabulary needs to be corrected throughout the manuscript to put the paper in a better context and less misleading interpretations. Clearly no epigenetic element, including chromatin structure acts independently, such that DNA methylation, ncRNA and histone modifications are required to get any defined chromatin structure. This needs to be thoroughly revised throughout the manuscript.

Chromatin “structure” is a hypothetical model developed based on interpretation of various data for understanding genomic phenomena. Chromatin “contacts” or “interactions” are direct outcomes of assays based on the proximity ligation technique, which is also known as chromosome conformation capture. Hi-C sequencing is a typical application of the proximal ligation technique, and its outcome can only describe chromatin contacts or chromatin

interactions. Ironically, our manuscript submitted to another Nature journal received a review critique that chromatin “structure” is rather inadequate for our manuscript because the data presented in it shows chromatin “contacts or interactions.” We prefer sticking with the more precise terminology (i.e., contacts/interactions) and have adjusted the discussion to make our position clear and address the comments of this reviewer.

5) Although the manuscript requires a better vocabulary, context, and inclusion of all epigenetic factors to be accurate and non-biased, this reviewer feels the manuscript can be thoroughly revised to present the data obtained and put into a thoughtful and more accurate context.

We have made revisions throughout the manuscript to remove any phrases that may be interpreted as our bias while still indicating that we favor a model in which the new chromatin contacts in and around the *Id* gene play a prominent role in the observed phenotypes.

Reviewer #3 (Remarks to the Author)

Chang and colleagues report in this manuscript that exposure to tributyltin (TBT) causes heritable changes in chromatin interactions (CIs) in PGCs in F1s, which can be stably maintained in PGCs and the liver of the subsequent unexposed F3 generation. In the only DCI identified to be persistently maintained across three generations, the downregulation of *Ide* in the liver was shown to correlate with the hyperglycemic and hyperinsulinemic phenotype in F3 mice. The authors concluded that CI changes may represent a molecular mechanism underlying the transmission of the transgenerational predisposition to obesity due to gestational exposure to an environmental obesogen. While the idea that changes in chromatin configuration underlie transgenerational epigenetic inheritance is of great interest, it is not completely novel, and the data shown in this report are largely correlative.

General:

This study raises more questions than answers. For example, why was only one DCI maintained in F3 while 20 ICs were detected in F1 and F2?

In this study, we began with a high stringency selection criterion and found that only a single DCI was maintained in F1-F3 generation PGCs. When the stringency was lowered, or only F1 and F2 generations were considered, more DCIs became evident. Instead of presenting all such low-stringency DCIs and stopping there, in this study we decided to take a deeper look at the single DCI (formed around the *Ide* gene) that survived with the most strict stringency and are relevant to the observed phenotypes.

Why are only males affected?

This is an important question and the subject of the recent, successful 5-year renewal of the NIH R01 that supports this work.

Why were the DCIs detected in PGCs maintained in the liver but not in other organs?

The most stringently detected DCI in PGCs was in the gene encoding *Ide*, and *Ide* mRNA was downregulated in liver in two independent experiments in F2-F4 generations. Based on these observations and given the *Ide*-relevant metabolic disorders we observed (hyperinsulinemia, hyperglycemia, hyperleptinemia) we argue that the emphasis on *Ide* is justified. We do not rule out a possible role for other genes, but down-regulation of hepatic *Ide* can readily explain the phenotype observed. It is possible that the DCIs may also be maintained in other organs, but our current analysis focused on the liver because it is where the action of insulin degrading enzyme, plays the most critical role in insulin clearance.

Are other genes in the only DCI dysregulated? If so, what are the contributions of those genes?

Expression of *Hhex* and *Kif11* are up-regulated in males as a consequence of the new DCI in the *Ide* region. Both of these loci, along with *Ide*, have been firmly associated with type 2 diabetes in human cohorts by many independent studies. However, as of today nothing is known about their function in obesity or diabetes. As noted above, impairment of insulin clearance should be sufficient to produce the physiological phenotype we observed and reproduced.

Are those DCIs detected in PGCs maintained in sperm?

We did not analyze sperm in this experiment and do not have any remaining samples that could be analyzed.

This reviewer understands that these may be topics for future studies. The large gaps between DCIs and the observed phenotype leave the study purely correlative and hard to believe.

If we can manipulate TAD formation around the *Ide* locus in live mice, we would be able to present strong evidence of the causation. Such studies – including inducible CRISPRa/CRISPRi manipulations of the *Ide*-associating TADs and direct regulation of the affected genes – are ongoing in our laboratories. However, as outcomes of such experiments will be available only two or three years later, the causative evidence has to be presented as a separate study in the future – not to be covered in the current manuscript. However, the fact that new contacts were introduced into the *Ide* gene and detected in this experiment, resulting in down-regulation of hepatic *Ide* expression and impaired insulin clearance, together with the observation that *Ide* is similarly downregulated in liver remaining from a previous transgenerational experiment (T3)(Extended data figure 6d) provides strong evidence that the finding is more than an accidental correlation. We argue that the reproducibility of the *Ide* down-regulation phenotype in 2 independent transgenerational experiments, coupled with the consistent change in chromatin in and around the *Ide* gene is powerful validation that our model is likely to be correct .

Specifics:

Validation of CI formation in the *Ide* gene was conducted using ChIP-qPCR, which can be biased. Wouldn't it be better to use ChIP-seq? ChIP-seq would allow for the validation of the Hi-C data as well.

The median size of TADs and subTADs in mammalian cells is approximately one megabase and 100-200 kb, respectively. Due to technical limitations, especially the number of mouse PGCs collectable from E13.5 embryonic gonads, the Hi-C data currently presented in our study has a resolution sufficient for detection of (sub)TADs but not lower-scale chromatin contact events. In contrast, ChIP-seq typically detects protein-DNA interactions with less than 1 kb resolution, and ChIP-PCR can target precise protein binding sequences with less than a 100 bp window. Due to these technical differences in the targeting windows, ChIP-qPCR provides an opportunity to detect weak protein-DNA interactions at specific loci that may not be readily detectable by ChIP-seq.

Following this review comment, we performed ChIP-seq experiments for CTCF interactions with genomic DNA in livers from F3 DMSO and TBT group males after F0 dam exposures to TBT or DMSO (no additional PGC samples from this experiment were available for submission of this revision). Strikingly, we observed strong differences in CTCF binding sites in the close proximities of the differential TAD boundaries that were estimated by using our Hi-C data, agreeing with the reviewer's suggestion that the ChIP-seq data would allow the validation of the Hi-C data. We appreciate the reviewer for this constructive suggestion and added outcomes of the CTCF ChIP-seq data to this revision as Extended data Fig. 6 and Extended data Fig. 7).

The title and the text all mentioned obesity as the phenotype, but the data showed no significant changes in either body weight or composition. The changes detected reflect glucose metabolic disorders.

Figure 1a,b show that the body composition of HFD challenged male mice changed as a result of the ancestral TBT exposure. We have also published this male-specific increase in body fat phenotype in two other publications from previous transgenerational experiments (T2, Chamorro-Garcia et al, 2017) and T3 (Chamorro-Garcia et al, 2021).

Given that significant reprogramming occurs during the meiotic phase of spermatogenesis (NCB, 2023, 25:1520), some of the CI changes may be reset and thus disappear in sperm. Are those detected DCIs present in sperm as well?

We do not have sperm remaining from this experiment. However, each of the F3 animals analyzed in this manuscript is derived from a different F0 male and F0 female from every other animal. That is, pedigree analysis supplied in response to reviewer 4 clearly demonstrates that the 16 F3 males and 16 F3 females in the DMSO and TBT groups were derived from different F0, F1 and F2 matings. How environmentally induced epigenetic changes survive the genomic reprogramming during gametogenesis is a central question in the field of TEI. Our current study has anchored early (PGCs) and late (adult phenotypes) points through epigenetic changes at a single specific locus. Future studies will be needed to connect details of these two points, including genomic analysis of gametes. The current study does not include analyses of gametes although we did include PGCs from F1, F2, and F3 generations in both sexes.

Reviewer #4 (Remarks to the Author):

This manuscript describes experiments to examine whether 3-D genome interactions as detected by HiC could potentially be involved, or associated with multigenerational phenotypes observed in offspring of mice treated with TBT. Overall, the manuscript is very clearly written considering the challenge often associated with communicating inter/transgenerational literature. The phenotyping data appear well done. Overall, the data are correlative without a test of causality or replication.

We agree that causality has not been tested, however we note that the down-regulation of *Ide* in TBT group animals has been replicated in two independent experiments. Replication of such a specific phenotype, down-regulation of *Ide* expression associated with diet-dependent hyperinsulinemia, hyperleptinemia and increased fat mass, supports our model.

The phenotypes themselves raise no concerns and are in keeping with obesity-prone developmental programming, and therefore are important. The genomic data and experimental design raise a number of key questions that leave the impact of the work unclear. These issues are of high importance given that the entire field currently suffers from the legacy of such critiques. It is therefore important at this level of publication that the experiments, data and design rise up to a definitive level. The authors are lauded for the work invested. My significant concerns in order they arose:

- Can the authors please show a pedigree of the experiment including all animals and color code perhaps which animals were used for what experiments?

This has been provided as Extended data Fig. 17. Every single F3 male analyzed was the product of different F0, F1 and F2 matings.

It is unclear why so many mice were originally ordered

The number of mice used was based on a power analysis aimed at detecting a 10% difference in body fat with sufficient statistical power. We also have a stringent selection criterion for litter size, requiring litters of 6-8 pups with at least two of each sex (one for phenotypic analysis and another for breeding the next generation).

and what the number of biological replicates is for the Hi-C. Is it an n=1 pooled from many?

The number of biological replicates for Hi-C is 5. That is, PGCs isolated from embryos of a single dam were pooled by sex and each pool was n=1. Hi-C was performed from 5 replicates (i.e., embryos from 5 different dams) and the data pooled by treatment, i.e., DMSO group males, DMSO group females, TBT group males, TBT group females.

Paternal variation appears not to have been eliminated by having an equal number of DMSO and TBT pregnancies derived from each male. Is this true?

Each pregnancy was derived from a single male as demonstrated by pedigree analysis (Extended data Fig. 17); therefore the reviewer's concern is addressed by our experimental design.

The fact that paternal intergenerational effects exist and that they have been suggested to potentially elicit CTCF-dependent, should be mentioned. Similarly, given that virgin/primiparous mothers lactate and behave differently than multi-parous mothers, were these all virgin crosses?

We are aware of the previous studies providing evidence that CTCF binds to genomic DNA in sperm and forms chromatin interactions therein. Following the reviewer's suggestion, we mentioned it in Discussion with a new citation as follows:

"There is also evidence that CTCF binds genomic DNA in sperm to form chromatin contacts, which may function as vehicles of paternally transmitted epigenetic inheritance (Gold, Jung, Corces 2018 JBC, PMC6130957)."

And

"Whether the observed, TBT-induced changes in chromatin contacts around the *Id4* gene in primordial germ cells are originated from altered CTCF binding in sperm, or such changes are preserved in the sperm genome, are interesting questions."

These were all virgin crosses which was intended to eliminate any confounding effects of multi-parous vs virgin dams.

- How are animals chosen? Are they the most median-like for their respective group (eg. the most average sex-specific body weight for the litter? Randomized sampling, when overall sample number is low, 'needlessly' includes outliers at a frequency that 'randomly' may impact one group more than another. This chance is reduced as sample number increases. Similarly, it provides a robust measure of control to experiments for a field that is often criticized for such design (eg. inter-/transgenerational effects).

Animals selected for breeding the next generation were randomly chosen from the entire group of animals which passed the litter size criterion (6-8 pups \geq 2 of each sex per litter) and after eliminating the smallest and largest remaining animals (outliers) at weaning. Since our experiments used individuals from n=16 litters for each treatment group and outliers were not included in the pool from which random animals were selected, there should not be any outlier effect that the reviewer noted.

- What is the rationale for the different times of HFD treatment in F2 and F3. Given the early timepoint associated with HFD in F2, does this mean the F3 are derived from F2 that also experienced HFD?

Littermates of the animals chosen for HFD analysis were used for breeding the next generation in every case; therefore, the animals chosen to breed the F2 and F3 generations were never exposed to HFD challenge. The choice of different times for the HFD experiment was based on an attempt to shorten the length of the experiment. While we typically used 12 weeks (See

Chamorro-Garcia et al, 2021) for the diet challenge, we shortened this to 5 weeks for this experiment based on another study of ours (Chamorro-Garcia et al, 2018) that only examined phenotypes in F1, While the diet challenge in the F2 generation was successful at 5 weeks (Figure 1a), it was not successful for the F3 generation (Extended data Figure 1); therefore we used reserved animals from the same group that had been maintained on a standard chow diet and subjected them to diet challenge at 17-weeks. This diet challenge was successful. We have standardized the diet challenge at 12-weeks in our subsequent experiments and this has proven to be reproducible.

- It sounds like the animals in Fig 1b were also exposed to HFD at 5w of age but this is not indicated on the graph. HFD is well known to have substantially different effects when applied early vs late, and HFD creates substantial metabolic effects irrespective of weight-gain.

The time of treatment with HFD is indicated in all parts of Figure 1. For the F3 animals in Fig. 1b,d it was 17 weeks as noted. Extended data Fig 1b shows the effect of treating F3 males with HFD beginning at 5 weeks and continuing through 11 weeks and is discussed in the text.

- The metrics used to conclude that the Hi-C profiles are globally similar with exception of these detected DCI's are insufficiently self-critiquing. There is no evidence of intra-condition variability against which to gauge inter-condition variability.

The intra-condition variability was far smaller than the inter-condition (DMSO vs TBT) variability. However, this observation needs to be interpreted carefully because comparisons between each single PGC samples have reduced sensitivity due to the smaller numbers of PGCs and deep sequencing reads and so less power to detect differential TADs. The almost completely clear background of the differential TAD detection within chromosome 19 indicates a very low level of noise derived from the intra-condition variability.

- X and Y chromosome mapping counts across samples should be shown given the pooling approach.

Prior to Hi-C analysis of PGCs, we verified by PCR that each pool was exclusively comprised of male or female embryos. We include a table in the revised manuscript showing X and Y chromosome counts, as requested (Extended data Fig. 16). The Y chromosome counts for males were well balanced among all samples, intra- or inter-condition categories. Female samples show very small but non-zero Y chromosome counts due to known strong similarities in the genomic DNA sequences between the X and Y chromosomes; these counts would not affect interpretation of the Hi-C data. The X chromosome counts were approximately twice greater in females than males, which is expected, and well-balanced intra- and inter-conditions.

- Embryonic sufficiency and phenotypes co-vary with the position of each embryo relative to blood supply, with those animals closest to the arterial supply being larger than those more distal. In animals that undergo natural birth, this information is lost and inaccessible. In animals that are taken at fetal stages however this variable can be controlled for. Was embryonic position accounted for?

While uterine position was recorded for embryos after they were sexed, since the PGCs from each litter were pooled by sex, these data were no longer available.

- One alternate explanation for constellation of results specifically at the *Ide* locus is a genomic structural rearrangement. Some translocations/inversions, and in particular genomic duplications, can trigger the same pattern of Hi-C results and longevity of phenotypic changes. This reviewer is not an expert in SDquest; is the tool robust enough to rule out that a subset of animals within a pooled replicate contain an SV?

Although SDquest has a historical significance, this 6-year-old software has never been updated since its original version (ver. 0.1). Several newer software tools have improved our understanding about large events such as translocation, inversion, or duplication, which are readily accessible through publicly accessible databases. We examined these databases and have not found any evidence of large genomic events that would affect interpretation of our chromatin interaction analysis. Whereas the dTADs reported in our current study span 36.5-37.5 Mb coordinates in chromosome 19, the *Cyp2c* cluster (apparently reflecting gene duplication events) is found at 39.0-40.5 Mb. However, the 1.5-Mbp distance between the *Ide-Kif11-Hhex* region and the apparent nearest gene duplication event at the *Cyp2c* cluster sufficiently segregates the former from the latter with no evidence of significant physical or functional links between them.

Even a minor fraction of such events should yield a robust and reproducible signal in HiC (and phenotypes). This could be approached by examining even two independent biological replicates from distinct families/pedigrees (assuming this is not a striking hotspot event). Again, are there replicates (from distinct pedigrees) that could help rule this out? alternatively, if the authors still have DNA from other tissues of the specific F2 and F3 animals involved, a methylation array analysis, WES, DNA microarrays could all be used to assess copy number across individual animals to rule this in or out presumably without enormous costs (or even in technical replicates of the pooled samples).

We examined 5 biological replicates, all from separate pedigrees. Therefore, the possibilities raised here are ruled out.

- Do the authors discount potential roles for the many other DCI's observed that didn't survive the approach?

We cannot not positively exclude possible roles of the DCIs that were eliminated by our stringent filtering criteria in the observed transgenerational phenotypes. Such DCIs may be examined in separate studies in the future. However, our data support the prominent role of the single DCI around the *Ide* gene (encoding Insulin degrading enzyme) in the transgenerational inheritance of the diet-induced, male-specific increases in WAT depot size, hyperinsulinemia, hyperglycemia and hyperleptinemia.

- Given the issues above, the fairly strong implications about causality should be toned down (or eliminated until causality is tested) and replaced with alternate explanations / discussions of mechanisms through which these changes would be expected to be repeatedly induced and

how they might survive (or be saved/protected) through epigenome erasure and development. Do the authors suggest that these changes are maintained through toti- / pluripotency? Do they have corroborative evidence for instance using FISH that these contacts are indeed increased?

We have toned down the discussion about causality, although another explanation for why hepatic *Ide* expression is down-regulated in 2 independent experiments is not obvious. The most parsimonious explanation is that the new contact introduced into *Ide* results in its down-regulation and the phenotypes we have observed here. Our future experiments, supported by a new 5-year R01 award aim to test causality by manipulating the structure of the *Ide* region.

- Early post-weaning period is a phase of life still considered development and overlapping with substantial endocrine fluctuations, final maturation of the gonads, as well as puberty/menarchy. HFD appears to have been administered during this time period in one set animals (F2, if I understand the methods correctly). HFD is well known to impact 5-8 w old mice very differently than adult mice. Also, mice immediately post-weaning have just moved from mixed milk/chow to chow-only diet and their metabolic tissues are undergoing a related wave of environmentally triggered terminal maturation and their microbiome is also highly variable between animals. Was there a rationale I am missing to start an intervention given the high degree of unmeasured variability and developmental rearrangement that is happening in this time window? With all due respect, this is truly unfortunate for an experiment focused on variation/plasticity. All figures/text should make clear that F2 underwent 2 interventions. The authors are applauded for the transparency in the report.

While we agree with the reviewer on this point regarding the possibly different effects of HFD challenge between weeks 5-11 vs 17-22, we also note that the physiological effects of HFD challenge are remarkably similar in F2 and F3 animals (Figure 4, Extended data Figures 8-11). Therefore, despite the unfortunately different timing of the diet challenge in F2 and F3 generations, the physiological effects are quite similar.

Minor

- No DMSO/TBT legend on Ext data 10, unless I missed it.

We thank the reviewer for pointing out this omission, which has been corrected in the revised figure legend for what is now Extended data Fig. 9

Reviewer #2 (Remarks to the Author):

The previous revisions have improved the manuscript and I appreciate the comments provided in the revision comments section. Although I feel the manuscript should be published, the following comments are provided to allow a further revision to improve the manuscript and put it into more accurate and less biased context. I feel this would improve the manuscript and provide a less biased presentation for the reader, but this is simply a suggestion for the author, and should not prevent publication.

1) Abstract, the controversial comment in the Abstract is accurate, but this is not due to the extensive literature on the topic of transgenerational inheritance, but this is primarily due to the concept of genetic determinism, which is predominant in today's science. Therefore, a discussion of controversy without mention of this issue is misleading.

In response to this comment, we added the following sentences to the third paragraph in Discussion:

“The existence of TEI in mammals remains controversial primary due to the currently predominant notion that changes in DNA nucleotide sequence are the exclusive basis of inheritance of potentially acquired traits. Another source of the controversy is the wide variety of epigenetic mechanisms that have been proposed to explain mammalian TEI without describing how more than a single type of molecular epigenetic events can be involved.”

2) New literature on the lack of DNA methylation erasure is not presented in regards to the comment DNA methylation is erased during embryogenesis. Observations now indicate the lower density DNA methylation that constitutes the majority of DNA methylation is not erased in the primordial germ cell, as occur with high CpG density sites. The transgenerational literature on ncRNA is also not referenced. The reviews listed are primarily older, so inclusion of new literature would be useful, lines 50-64.

We appreciate the reviewer's suggestion and have added a discussion on recent findings regarding DNA methylation erasure in embryogenesis. Recent studies (e.g., Ben Maamar et al, 2023) indicated that while high-density CpG DNA methylation is erased, lower-density methylation is retained, potentially contributing to transgenerational epigenetic inheritance. Additionally, accumulating evidence supports the role of small non-coding RNAs (ncRNAs) in carrying epigenetic information across generations. We have added the following to the manuscript at Line 54:

“DNA methylation is erased genome-wide twice each generation in mammals¹³; although emerging evidence suggests that lower-density DNA methylation may persist in primordial germ cells, contributing to transgenerational inheritance¹⁴.”

In response to the ncRNA point, we added the following to the manuscript from line 62 (new text from line 64-66):

“It remains uncertain how expression of non-coding RNAs, such as those that were reported in F1 sperm and seminal fluid¹⁶ could be transmitted to multiple future generations in mammals, although, small non-coding RNAs (ncRNAs) have been linked with epigenetic transgenerational inheritance¹⁷.”

3) Stable generational changes for DNA methylation and ncRNA have been shown in the literature, but this literature is ignored and states TADs only evidence, lines 256-264. Literature clearly demonstrates corresponding DNA methylation and histone modifications are required for chromatin structure and TADs to form, but this is not clarified in the Abstract or Discussion sections.

We addressed this concern as noted above.

4) The bias to TADs or chromatin structure needs to be reduced. The clarification all epigenetic mechanisms are integrated and will be involved in epigenetic transgenerational inheritance needs to be clarified. The literature on this topic is not presented.

We recognize the need to present a more integrated view of epigenetic mechanisms in transgenerational inheritance. It is our model that DNA methylation, histone modification and ncRNA expression may be effectors of transgenerational inheritance that are downstream of changes in higher-order chromatin structure that carry the phenotype. This is discussed from lines 268-277.

Reviewer #4 (Remarks to the Author):

This reviewer appreciates the author's response and efforts. They have addressed most of the concerns. There are some remaining issues below, and with apologies, one issue of high relevance that must be brought up again.

Revisiting copy number variation (CNV)

Yesterday when presenting my own developmental plasticity work, a postdoc in the crowd asked if I was aware of the following paper by Watkins-Chow and Pavan - <https://genome.cshlp.org/content/18/1/60.full>. I was not. In reading it today on the flight I see it shows what appears to be convincing evidence of a dual signal CNV in Jax C57BL6J animals. The dual CNV signal precisely overlaps the signal identified in the manuscript under review here (i.e., two likely-coupled CNV signals that match exactly the two Hi-C signals at *Ide* and *Fgf3*, one shorter one over *Fgf3*, and the second, slightly longer one, directly over *Ide*). Watkins/Pavan describe the CNV as being present at frequencies consistent with the possibility that they underpin the transgenerational effects described here. Presumably, the origin of the problem was that Jax maintains extremely large colonies that make maintaining 'isogenicity' near impossible when generating and maintaining their colonies. This reviewer is unaware of what Jax has done since to remedy that situation. We will also be reaching out to Jax for clarity. On quick survey, I cannot find evidence that they have remedied the situation, and the postdoc who informed me of this indicated that they personally detected the CNV in many mice, ordered from Jax in 2020. These data raise significant concerns of a genetic underpinning for the current manuscript that have to do with the isogenicity/fidelity of C57BL6J line itself.

Response:

We thank the reviewer for introducing us to the study of Watkins-Chow and Pavan (WC/P), which was published in 2008 to describe the two CNVs around the *Ide* gene in chromosome 19 of the C57BL/6J mice. However, to the extent we investigated, the existence of these CNVs has not been supported by subsequent studies despite the critical importance of this particular strain, which represents the GRCm mouse reference genome. The negative reports include a comprehensive study from the Jackson Laboratory scientists on the genome of C57BL/6J "Eve," which mentioned the WC/P study but showed no evidence of the *peri-Ide* CNVs in either the founder mouse of this strain or its contemporary offspring [Sarsani et al, 2019 PMC6553538]. Although we noticed a few studies on the *peri-Ide* CNVs published after the WC/P paper, in our opinion none of them convincingly reproduced the WC/P CNVs. Therefore, rather than a well-established fact, whether or not a CNV exists near *Ide* is controversial and certainly bears further study.

Because of the near impossibility of coincidence, particularly the exact overlap of both CNV signal-elements with the Hi-C signals identified here in this work, this begs a serious re-examination of the possibility that CNV variation underpins the incomplete penetrance.

Response:

Perhaps we were not clear enough in our description of the phenotypes. The transgenerational predisposition to diet-induced obesity, decreased *Ide* expression and elevated insulin levels is 100% reproducible across all batches of male offspring of TBT-treated pregnant F0 dams and 100% reproducible (absent) across all of female offspring. Moreover, we have reproduced this male-specific predisposition to diet-induced obesity in 5 independent transgenerational studies using C57BL/6J conducted at UCI [Chamorro-Garcia et al, 2017, Chamorro-Garcia et al, 2019, Chang et al (current manuscript), Chang et al (under review at Endocrinology, Chang et al, in preparation)] and another conducted at MGH using OG2 mice (Tg(Pou5f1-EGFP)2Mnn) (Shioda et al, 2022). The probability of an incompletely penetrant CNV in *Ide/Fgfbp3* underlying the male-specific phenotype (and not in their female littermates) in 5 independent experiments is vanishingly small, particularly since Sarsani et al did not observe this CNV in any C57BL/6J mice in the Jackson Labs colony. See also the discussion below.

This reviewer would recommend replicating some or all of the techniques used by Watkins/Pavan to test the issue. This must of course be done on samples from individual mice (not pools, and ideally those used in the study), and from tissues that do not contain large variation in polyploidy that may mask the signal (i.e., definitely not liver; ideally haploid sperm... I guess?... I am not a CNV expert, yet). The authors should also consider long-read sequencing as well as using a read mapping/alignment strategy that allows for chimeric-read mapping.

Regardless, given the essentially perfect overlap with the reported CNV signals, documenting these validation results and clearly relating them to how/when the C57BL6J colony used here was distinguished from Jax (or relative to what Jax has done, if anything, about the issue) is necessary. As a scientist, I feel this is completely inappropriate for Jax not to highlight boldly in their ordering system if it is still happening. We will also begin carefully looking into this in our lab. I feel for the authors and in many ways hate to have to be the one that brings it up.

Response:

We agree with the reviewer that the regions of the peri-*Ide* CNVs detected by array CGH in the WC/P paper and the differential chromatin contacts detected in our current study by Hi-C are practically identical. Array CGH is not free of artifacts, which can be caused by wet lab procedures or computational analytic algorithms. Unfortunately, in the WC/P study the wet lab procedures were outsourced to a commercial service provider (Nimblegen Systems). In the absence of information on the wet procedures, we are unable to replicate the WC/P study. For the dry analysis part, the WC/P study did not include *spatial normalization*, which is an important step that effectively removes false positive artifacts. Therefore, the array CGH data presented in the WC/P paper is not necessarily the gold standard. The chromatin contacts around the *Ide* gene could have been caused by anomalous DNA labeling, biased DNA amplification, or inadequate data normalization in the array CGH experiments, resulting in the false detection of CNV at the same location of the chromatin contacts.

We also agree that the presence of the CNVs described by Watkins-Chow and Pavan could influence the Hi-C outcomes we reported since our Hi-C study lacks the resolution to readily distinguish between new contacts present in the *Ide* region and increased copy number of the affected region. Although, it seems impossible that our phenotypes could be caused by the Watkins-Chow and Pavan-reported CNV, we recognizing the significance of this issue and attempted multiple approaches to assess whether CNVs could play a role in our findings.

To investigate potential CNV contributions, we:

1. **Attempted to remove possible effects of CNV in our Hi-C data.** To the best of our knowledge, the most widely accepted method to remove the effects of CNVs in Hi-C data is CAIC normalization (Servant et al, 2018, PMC6127900). We attempted to use this method to reduce any possible effects of CNVs in our Hi-C data. Unfortunately, this method broadly suppressed TAD detection throughout the genome in all samples, demonstrating that it was not suitable for the task by rendering the differential TAD (dTAD) analysis unreliable.
2. **Reviewed existing literature**, including the WC/P study, and considered their findings in the context of our results. While WC/P reported the presence of two small CNVs near the *Ide* locus, as noted above, a recent comprehensive study of the C57BL/6J genome by Sarsani et al, 2019 PMC6553538 found no evidence for any CNVs on chromosome 19 other than a small duplication between 4.816-4.817 MB, which is nowhere near to *Ide* (37.27-37.33 Mb), Therefore, whether or not a CNV exists near *Ide* is controversial.

3. **Considered our findings** in light of WC/P who reported that the CNV near to *Ide* led to **~1.5 fold elevation** of *Ide* expression in the tissues they evaluated (spleen and brain) and a 1.5-fold elevation of *Fgfbp3* in spleen but no changes in brain.
- a) In contrast to what would be expected by the additional copy(ies) of *Ide* were they present in our mice, we found **~1.5 fold decreased** expression of *Ide* in liver and no changes in other tissues analyzed in the paper (muscle and WAT) only in male animals from the TBT group. No changes were observed in females. We assessed *Ide* mRNA levels in brain and spleen to facilitate direct comparisons between our paper and WC/P and found no changes in *Ide* expression (Extended data Figure 21). Therefore, it is unlikely that our results could be explained by the CNVs described by WC/P, even if they existed in the 5 groups of animals we obtained from Jackson Labs over the past 10 years.
 - b) We note that the CTCF ChIP-qPCR shown in Fig 3 and Extended Data Fig 5 showed enhanced CTCF binding at some (B, C, F, G), but not all (no increased binding to A, D, E) of the known and predicted CTCF binding sites in the *Ide* gene from male TBT group mice. No increased CTCF binding was noted at any site in female liver DNA. This is inconsistent with what would be expected if the putative dTAD was the result of a CNV rather than being a bona fide dTAD.
 - c) We randomized all mice obtained from Jackson labs prior to starting the experiment and treated groups of 74 F0 females with either DMSO vehicle or TBT. Breeding of subsequent generations used groups of at least 61 (F1) and 69 (F2) females per treatment. While it might be remotely possible that only the TBT group females accidentally received mice with CNVs, it is inconceivable that only the male offspring of these mice in every generation received this autosomal CNV compared with their female littermates which showed no evidence of the dTADs or increased CTCF binding. Moreover, as noted above, the male-specific predisposition to increased fat mass and *Ide* down-regulation has been reproduced in multiple, independent experiments. Therefore, we cannot envision a scenario in which genetic inheritance of a CNV that is randomly distributed in the population drives this highly reproducible phenotype.

Ultimately, while our data do not absolutely rule out the potential involvement of CNVs at the *Ide* and *Fgfbp3* loci in the observed transgenerational phenotype, the data provide strong evidence that it is unlikely that the CNVs described by WC/P, if they exist in our animals, could be driving the changes in *Ide* expression and other metabolic phenotypes that are found exclusively in male descendants of TBT-treated F0 pregnant

dams. Given these constraints, we have expanded the discussion in the manuscript (lines 325-351) regarding this issue and highlight it as an area for future research.

In the case that the above issue turns out not to be an issue, the following points remain:

From last review Point 4 that highlighted potential relevance of the numerous genes within the TAD:

The authors responded: "To address the reviewer's comment, we now include qPCR analysis of the expression of **Fgfbp3**, **Btaf1**, **Cpeb3**, and **March5** (Extended data Figs. 19, 20). With the exception of a slight increase in hepatic expression of **Cpeb3** in males (~1.3-fold) and a small decrease in expression of **Btaf1** in gWAT of F3 males, there were no significant changes observed in the expression of the indicated genes. Therefore, the hypothesis that these genes are equally affected as **Ide** or equally causal to the multiple evidenced phenotypes is not supported."

The authors are commended for their work. Because it is beyond the scope of this work to rule out involvement of the TAD-embedded candidate genes across all relevant metabolic tissues and cell types, something like the following idea should simply be added to the results and discussion:

'Our findings are consistent with a mechanism that involves altered hepatic **Ide** and **Cpeb3**. The data do not rule out involvement of other genes within the TAD in other metabolic tissues, for instance through central (e.g., hypothalamic) or immune regulation. The Watkins/Pavan paper actually highlights the earlier point; brain and spleen appear to be relevant sites of altered expression in those CNV-varying mice.'

Response:

We appreciate the reviewer's insight regarding the broader involvement of TAD-embedded genes. In response, we have updated the discussion section of the manuscript to include the following (lines 298-304):

" Our findings are consistent with a mechanism involving altered hepatic **Ide** expression and the fact that the observed downregulation of **Ide** phenocopies the effects of **Ide** loss-of-function supports a central role for **Ide** in the phenotypes observed. However, these results do not rule out the potential contributions of **Cpeb3** or other TAD-embedded genes, in other metabolic tissues. Future studies are warranted to investigate the role of these genes in metabolic processes and their potential interactions with environmental exposures."

This addition highlights the need for further exploration while remaining within the scope of our current dataset.

From last review Point 5:

The authors responded: “In response to this comment, we amended the title to ‘Heritable changes in chromatin contacts linked to transgenerational susceptibility to diet-induced obesity.’ The major change observed is elevation of plasma insulin as a result of decreased insulin clearance, particularly in response to increased dietary fat. Therefore, we believe that the revised title is descriptive of what the manuscript presents.”

The authors state themselves that the major change is hyperinsulinemia and reduced insulin clearance, so why not put that in the title (“susceptibility to diet-induced insulin dysregulation”)? Those are the most substantial phenotypes; they ARE relevant to metabolic syndrome; they ARE metabolic sequelae of HFD; and they are NOT obesity (which as indicated has a really minimal phenotype). This reviewer would still recommend a more objective/accurate title.

Response:

We agree with the reviewer that hyperinsulinemia and reduced insulin clearance are critical, new findings of this study and that they are, in fact, present prior to HFD exposure, although, the insulin effects are strongly exacerbated by HFD. However, we disagree that obesity is a “really minimal phenotype” and point out that we are not using a standard HFD (40-60% Kcal from fat) but rather a diet with modestly increased fat content (21.6% Kcal from fat compared with 13.1% in the control diet). To better reflect these findings, we have revised the title of the manuscript to:

“Heritable Changes in Chromatin Contacts Associated with Transgenerational Susceptibility to Diet-Induced Insulin Dysregulation and Obesity.”

This updated title acknowledges the metabolic phenotypes observed in the study while maintaining a connection to the broader obesity-related context which we and others have found in multiple experimental systems following TBT exposure

From the reviewer’s most recent communication to the Editors

“I’m writing with a short follow-up on the potential genetic heterogeneity problem with JAX-derived C57BL/6J mice that I raised in the manuscript above.

We have now confirmed that this is indeed a problem and are already quite sure this will impact the entire metabolic community and likely beyond. We have primer pairs that can genotype mice easily as either ‘extra copy’ or ‘reference’ genotype, and we are already observing phenotype associations now that we have an easy tool. **If you remember, the key difference is that extra copy animals are essentially over-expressors for *Ide*.** So

when you order C57BL/6J mice from JAX, they send you a mixture of wild-type and over-expressors and you don't know who is who.

I feel I should share these reagents and invite the authors of the above manuscript to be included in our independent report intended to make sure this is highlighted extremely clearly to the field, and to provide the solutions. The financial costs of this 'mistake' by JAX could be in the... hundreds? of millions of dollars, and their contribution to confusion, irreproducibility and variability in the literature is all but guaranteed at least for the metabolic community. We are interested in working with all the major publishing journal groups to make sure this is broadcast as clearly as possible once we are ready to report there."

Response:

We thank the reviewer for making the effort to evaluate their own mouse stocks and to create reagents that can distinguish mice as to 'reference genotype' or 'extra copy', that is, *Ide* over-expressors as noted in the highlighted text above. This is a great service to the field and we appreciate it very much.

As we stated in our comments re CNV variation above, it is extraordinarily unlikely that any CNV in *Ide* can underlie the phenotypes we reported in our manuscript. As specifically noted in item #3 of our response above, we reiterate that our phenotype is *Ide* **down-regulation**, rather than the over-expression reported by WC/P, that our phenotype is male-specific (no phenotype or increased chromatin contacts in females), and that only some, but not all CTCF binding sites in *Ide* show increased binding in the male (but not female) TBT group. Moreover, due to the breeding scheme and large number of animals used, we cannot envision a scenario in which randomly selected females and males (from a population randomly containing 'over-expressors') gives rise to *Ide* CNVs (on the autosome chr 19) only in males of the TBT group, but not their female littermates, or either sex in vehicle controls AND that this group of *Ide* 'over-expressors' could be driving the observed phenotype. We have reproduced the dTAD in *Ide* in an additional experiment and shown that hepatic *Ide* expression is down-regulated in four (4) independent transgenerational experiments over the past 8 years (two shown in the current manuscript, one recently published <https://doi.org/10.1210/endo/bqaf063>, and an unpublished experiment for which we could provide the data upon request).

We would be open to testing the reagents provided by reviewer 4 in our animals and also to participate in a manuscript discussing the impact of the 'extra copy' mice sourced from JAX since we would be very interested in knowing which of our mice, if any, are *Ide* 'over-expressors'. However, we strongly believe these future analyses on JAX mice should not delay publication of our current manuscript, which reports novel

and independently validated findings. We trust this is not Reviewer #4's intent or the editor's expectation.

NCOMMS-23-53640C

Follow-up Analysis Efforts: After submitting our previous revision, we continued working to further evaluate the possibility of CNV at the *Ide* locus. We designed and completed WGS experiments on selected remaining liver samples from the F1 cohort of the transgenerational experiment reported in our manuscript in a further attempt to ascertain whether CNVs could be observed in the *Ide* region.

Approach: Genomic DNA isolated from F1 male livers in the DMSO group (n = 4) and TBT group (n = 4) was subjected to whole genome sequencing (WGS). Each group consisted of animals with the first and second highest, and the first/second lowest, % body fat content determined by EchoMRI. Thus, the full range of the body fat phenotype was represented by these animals. Importantly, all eight animals subjected to WGS were from different litters sired by different males as we presented in Extended data Fig. 17. This ensured that no two mice shared the same parental pairing.

This design prevents confounding effects due to shared parental genotype or in utero environment. We next examined whether CNVs existed at the *Ide* locus and whether any detected CNVs could be associated with either TBT exposure or metabolic (% body fat) phenotype. First, gDNA was extracted from previously snap-frozen liver tissue and processed for Illumina paired-end sequencing which was performed at the UCI High Throughput Genomics Core facility. The raw reads were filtered for Q>20 to retain high-quality reads and aligned to the GRCm38/mm10 mouse reference genome sequence using the BWA-MEM algorithm. To assess genome-wide read distribution, we generated normalized bigWig coverage files using the RPGC (reads per genomic content) method implemented in DeepTools. Coverage at the *Ide* locus was visualized using the Integrative Genomics Viewer (IGV), with a fixed y-axis scale applied across all samples to enable direct comparison of read depth and uniformity.

Figure 1. Whole Genome Sequencing analysis demonstrates the absence of genomic DNA copy number variations around the *Ide* gene. Illumina libraries were synthesized from genomic DNA of F1 male livers after F0 exposures to DMSO or TBT and subjected to whole genome sequencing. Reads aligned in a 1,853 kb region around the *Ide* gene of the GRCm38/mm10 mouse genome reference sequence were summarized into normalized coverage files (bigWig format) and visualized with fixed scales for each panel for direct comparisons across tracks. (A) Equal amounts of reads were pooled for four DMSO-exposure libraries or four TBT-exposure libraries. (B) Individual libraries exposed to DMSO or TBT with highest or lowest levels of body fat ratio.

Results: This analysis revealed no detectable CNVs at the *Ide* locus in any of the eight sequenced F1 liver samples, irrespective of the treatment group or % body fat status (Figure 1). These findings demonstrate that the differential TAD boundaries and chromatin remodeling we reported at the *Ide* locus in the manuscript under consideration are not the result of the reported CNV near *Ide*.

Conclusion: This additional WGS analysis supports our original conclusion that the chromatin differences at the *Ide* locus reflect epigenetic, higher-order structural changes rather than genetic variation. We hope that these new data, together with the information presented in the most recent rebuttal resolves the reviewer's concern regarding the possibility of CNV confounding our findings.

Response to Reviewer 4

As recommended by reviewer 4, we have now genotyped all of the F2 and F3 mice in our experiment for presence of the CNV. We found no significant differences between Ide CNV copy number in TBT vs control males, or TBT vs. control females (Extended data Fig. 22). We also plotted expression of Ide mRNA as a function of Ide CNV and found that there was no relationship between Ide mRNA levels and Ide CNV (Extended data Fig 23). Rather, the only differences noted were between TBT group males and control males, where Ide mRNA levels were significantly lower in TBT group males than in controls, even at the same Ide copy number. We believe that these results indicate that the presence or inheritance of Ide CNVs are not responsible for transmitting the transgenerational phenotype.

Rebuttal to Reviewer 2

On Revision 3: Reviewer #2 Remarks to the Author:

The revised manuscript is unfortunately not improved, and the specifics are listed below. Question on existence and mechanism of epigenetic transgenerational inheritance. Instead of addressing the concerns, the authors have added an increased number of inaccurate comments and speculations. This includes the following:

Line 21 – Burgeoning evidence suggestion; Lines 23-25 – Existence unclear TEI; Line 27 – Unanswered questions; Lines 28-29 – Attempts but not successful; Line 42 – All other mechanisms not plausible; Lines 49-50 – Possible mechanisms not valid; Lines 53-54 – No previous literature support mechanism TEI; Lines 62-64 – ncRNA role uncertain.

>>> Reviewer #2 evaluated our manuscript Revision 2 stating, “I feel this [review comment] would improve the manuscript and provide a less biased presentation for the reader, but this is simply a suggestion for the author, and should not prevent publication.” Apparently, changes we made in Revision 3 did not go along with the intention of the reviewer, despite that we made significant changes in specific response to Reviewer 2’s suggestions. Because all concerns expressed of Reviewer #2 are about cognizance of the current fields of research, in this revision (Revision 4) we further moderated the language. Accordingly, Abstract (Lines 21-48) and the first paragraph of Introduction (Lines 49-72) have been extensively revised. However, we also note that this is indeed an area of controversy in the field (as reviewer 2 well knows).

The data and discussion of data in Results section in general is fine and well presented.

>>> We appreciate this supportive comment.

The Discussion section also presents derogatory comments on past research on the topic. Lines 268-269 – Past conclusions TEI not accurate; Lines 270-274 – Involving a variety of epigenetic mechanisms suggested not correct in past literature; Line 272 – A single type of epigenetics involved not correct.

>>> We revised the relevant parts of Discussion to tone down the presentation. We did not intend to, nor do we believe that we did present derogatory comments on past research on the topic. Rather we pointed out issues that continue to prevent TEI from being fully accepted by the Genetics community.

With hundreds of manuscripts on each of these topics, the authors discount all previous science and research on epigenetic inheritance to suggest this study is the only one valid or important. Suggests interpretation all previous studies is wrong in broader perspective.

Therefore, the authors neither accept the past literature and for the most part disregard the past 25 years of research. For example, they disregard ncRNA mediated DNA methylation or DNA methylation mediated chromatin structure and the role of these processes in the formation of TADs, which is the focus of their study.

They also disregard the literature that all genetic mutations, like CNVs, have a precursor epigenetic element. For CNVs, the removal of the DNA methylation promotes the CNV expansion, or ability of transposable element to move, or role ncRNA in methylation, or role methylation in the genetic point mutations like C-T conversions. The manuscript suggests a lack of any past literature and integration of epigenetics and genetics and role of the different epigenetic components in genome biology, including chromatin structure.

>>> The length limitation imposed by the journal prohibits us from presenting comprehensive discussion on the past achievements in the relevant fields of research. It is not our intention to deny all preceding studies and instead promote our hypothesis as the sole correct one and we do not believe that this is an accurate interpretation of our presentation. Rather, we pointed out deficiencies

in the published data and interpretation, most of which have prevented TEI from becoming widely accepted by the Genetics community.

We believe that our current revisions in Abstract, Introduction, and Discussion address concerns of the reviewer.

On Revision 2: Reviewer #2 Remarks to the Author: The previous revisions have improved the manuscript and I appreciate the comments provided in the revision comments section. Although I feel the manuscript should be published, the following comments are provided to allow a further revision to improve the manuscript and put it into more accurate and less biased context. I feel this would improve the manuscript and provide a less biased presentation for the reader, but this is simply a suggestion for the author, and should not prevent publication.

1) Abstract, the controversial comment in the Abstract is accurate, but this is not due to the extensive literature on the topic of transgenerational inheritance, but this is primarily due to the concept of genetic determinism, which is predominant in today's science. Therefore, a discussion of controversy without mention of this issue is misleading.

>>> In response to this comment, we added the following sentences to the third paragraph in Discussion:
“The existence of TEI in mammals remains controversial primary due to the currently predominant notion that changes in DNA nucleotide sequence are the exclusive basis of inheritance of potentially acquired traits. Another source of the controversy is the wide variety of epigenetic mechanisms that have been proposed to explain mammalian TEI without describing how more than a single type of molecular epigenetic events can be involved.”

2) New literature on the lack of DNA methylation erasure is not presented in regards to the comment DNA methylation is erased during embryogenesis. Observations now indicate the lower density DNA methylation that constitutes the majority of DNA methylation is not erased in the primordial germ cell, as occur with high CpG density sites. The transgenerational literature on ncRNA is also not referenced. The reviews listed are primarily older, so inclusion of new literature would be useful, lines 50-64.

>>> We appreciate the reviewer's suggestion and have added a discussion on recent findings regarding DNA methylation erasure in embryogenesis. Recent studies (e.g., Ben Maamar et al, 2023) indicated that while high-density CpG DNA methylation is erased, lower-density methylation is retained, potentially contributing to transgenerational epigenetic inheritance. Additionally, accumulating evidence supports the role of small non-coding RNAs (ncRNAs) in carrying epigenetic information across generations. We have added the following to the manuscript at Line 54:
“DNA methylation is erased genome-wide twice each generation in mammals¹³; although emerging evidence suggests that lower-density DNA methylation may persist in primordial germ cells, contributing to transgenerational inheritance¹⁴.”
In response to the ncRNA point, we added the following to the manuscript from line 62 (new text from line 64-66):
“It remains uncertain how expression of non-coding RNAs, such as those that were reported in F1 sperm and seminal fluid¹⁶ could be transmitted to multiple future generations in mammals, although, small non-coding RNAs (ncRNAs) have been linked with epigenetic transgenerational inheritance¹⁷.”

3) Stable generational changes for DNA methylation and ncRNA have been shown in the literature, but this literature is ignored and states TADs only evidence, lines 256-264. Literature clearly demonstrates corresponding DNA methylation and histone modifications are required for chromatin structure and TADs to form, but this is not clarified in the Abstract or Discussion sections.

>>> We addressed this concern as noted above.

4) The bias to TADs or chromatin structure needs to be reduced. The clarification all epigenetic mechanisms are integrated and will be involved in epigenetic transgenerational inheritance needs to be clarified. The literature on this topic is not presented.

>>> We recognize the need to present a more integrated view of epigenetic mechanisms in transgenerational inheritance. It is our model that DNA methylation, histone modification and ncRNA expression may be effectors of transgenerational inheritance that are downstream of changes in higher-order chromatin structure that carry the phenotype. This is discussed from lines 268-277.